# Pangolin-inspired untethered magnetic robot for on-demand biomedical heating applications

Ren Hao Soon [1,2,6], Zhen Yin [1,3,4,6], Metin Alp Dogan [1], Nihal Olcay Dogan [1,2], Mehmet Efe Tiryaki [1,2], Alp Can Karacakol [1], Asli Aydin [1], Pouria Esmaeili-Dokht [1] & Metin Sitti [1,2,5] ✉

Untethered magnetic miniature soft robots capable of accessing hard-to-reach regions can enable safe, disruptive, and minimally invasive medical procedures. However, the soft body limits the integration of non-magnetic external stimuli sources on the robot, thereby restricting the functionalities of such robots. One such functionality is localised heat generation, which requires solid metallic materials for increased efficiency. Yet, using these materials compromises the compliance and safety of using soft robots. To overcome these competing requirements, we propose a pangolin-inspired bi-layered soft robot design. We show that the reported design achieves heating > 70 °C at large distances > 5 cm within a short period of time <30 s, allowing users to realise on-demand localised heating in tandem with shape-morphing capabilities. We demonstrate advanced robotic functionalities, such as selective cargo release, in situ demagnetisation, hyperthermia and mitigation of bleeding, on tissue phantoms and ex vivo tissues.

Untethered miniature robots can harvest energy made available to them remotely, such as light, magnetic, or acoustic energy, and convert them to other forms of energy, such as mechanical deformation[1,2]. Magnetic actuation has emerged as a promising method for robots in biomedical applications due to the magnetic field's ability to penetrate human tissues safely[3,4]. Advancements in this field have enabled these robots to navigate precisely to a target site on land[5,6] or in a fluid-filled confined environment[7,8], while carrying payloads, such as drugs[9–11], genes[12], hydrogel structures[13,14] and even patient cells, for therapeutic applications[15,16]. Furthermore, new developments in this field have enabled these robots to adaptively select their locomotion modes depending on the ambient temperature[17,18] and achieve anchoring in both tubular[19] and highly unstructured three-dimensional (3D) surfaces[20]. Despite these advances, the use of miniature magnetic robots in clinical applications is still limited because mainly one form of interaction, namely the mechanical interaction between the robot and the environment for locomotion or cargo delivery, is typically utilised.

In contrast, other forms of interaction between the robot and the environment are required in many biomedical applications. Heat energy, in particular, is a form of energy that is frequently used in common medical procedures, such as devitalisation, coagulation, and cutting, making it a desirable function for untethered robots to possess (Fig. 1A). There exist multiple methods to achieve remote heating, such as thermochemical, acoustics, photothermal and magnetic fields. Thermochemical methods are highly precise as the reagents are typically injected directly to the site and the products can usually be safely and easily excreted by the body[21–23]. Even so, the method is invasive and the heat produced cannot be easily localised. The reaction and the subsequent flow of reagents are highly dependent on environmental conditions, which cannot be easily controlled or predicted. Focused ultrasound is a non-invasive method for depositing energy on a target

[1]Physical Intelligence Department, Max Planck Institute for Intelligent Systems, 70569 Stuttgart, Germany. [2]Institute for Biomedical Engineering, ETH Zürich, 8092 Zürich, Switzerland. [3]Department of Control Science and Engineering, Tongji University, Shanghai, China. [4]Frontiers Science Center for Intelligent Autonomous Systems, Shanghai, China. [5]School of Medicine and College of Engineering, Koç University, 34450 Istanbul, Turkey. [6]These authors contributed equally: Ren Hao Soon, Zhen Yin. ✉e-mail: sitti@is.mpg.de

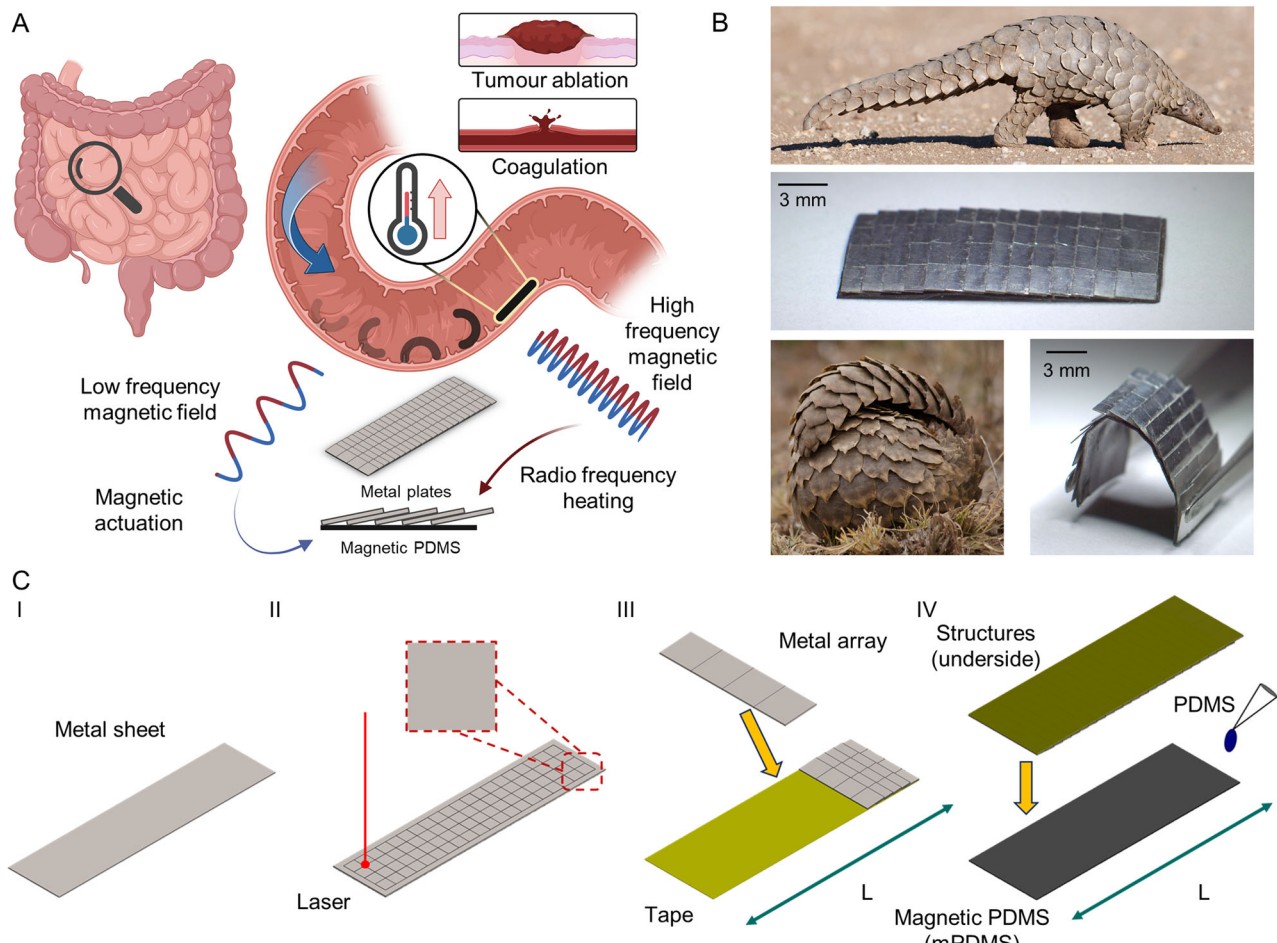

**Fig. 1 | Proposed pangolin-inspired RF heating mechanism for untethered magnetic robots. A** Conceptual illustration of the pangolin-inspired robot operating in the small intestine. The robot is actuated with a low frequency magnetic field to the target location. Application of a high frequency magnetic field results in Joule heating of the metal plates. The heat energy can then be used to interact with the environment. Inset on the right shows potential medical conditions in which a miniature untethered magnetic robot with heating capabilities would have utility. Figure created with biorender.com. **B** Armour on pangolins consists of individual overlapping hard keratin scales attached to the body. This design allows for rigid structures to be attached for protection without compromising on locomotion. Scaled robot inspired by this overlapping design is shown on the right. Images of pangolins used under licence from Shutterstock. **C** Fabrication procedure. (I) The metal of correct thickness and material is selected. (II) The sheet is laser cut to form the scaled patterns. (III) The metal arrays are detached and assembled on a tape. (IV) The assembled structures on the tape are bonded to the magnetic PDMS with PDMS before the tape is removed.

spot and not surrounding tissues[24–26]. However, ultrasonic waves are unable to pass through high-density materials, such as bones or if there is an air-liquid interface. In the same vein, photothermal methods also offer precise heating but only have a maximum penetration depth of 1-2 cm under the skin[27–29]. This limits their applications to outer regions of the human body unless a light source is brought inside the body. As such, magnetic methods are best suited to realise controllable, on-demand, non-invasive, and targeted remote heating deep inside the human body.

Remote heating can be achieved using alternating magnetic fields or radio-frequency (RF) fields via two mechanisms: Joule heating or hysteresis losses[30]. Although exploiting magnetic hysteresis for remote heating has been extensively studied and utilised for robotic[31–33] and hyperthermia applications[34–36], magnetic hysteresis dissipates significantly less heat as compared to Joule heating. In fact, magnetic hysteresis remains a more popular method than Joule heating because the magnetic nanoparticles used simultaneously fulfil other functions of the robot and/or heating is not the primary output of these robots. For instance, the use of magnetic nanoparticles to achieve heating via magnetic hysteresis allows the material of the robot to still remain soft for large changes in volumes[32,33] or because these nanoparticles have

desired chemical or biological characteristics[35,37]. While there is a third method of achieving remote heating with RF fields through the use of RF resonant circuits[38–40], we do not consider these methods because the RF coupling efficiency is less robust and is therefore, unable to produce reliable, controllable remote heating. The coupling efficiency is highly dependent on environmental conditions, such as the ambient temperature, ionic content and concentration and even the physical deformation of the robot itself to name a few[41] – all of which are highly difficult to predict, especially in a highly heterogeneous environment, such as inside the human body[42].

As demonstrated in this work, rigid metallic body parts should be used for remote heating to ensure that the electrical conductivity and geometrical properties remain constant and stable for enhanced and reliable remote heating via Joule heating. However, the use of rigid materials inherently restricts and compromises the compliance of such untethered soft robots, taking away a key advantage of having a soft body with diverse shape programming capability. To address this inherent trade-off between effective remote heating at long distances and compliance, we observed how pangolins in nature could still achieve flexible and unencumbered motion despite having keratin scales,

which are orders of magnitudes harder and stiffer than the underlying tissue layers, simply by organising the keratin scales into an overlapping structure[43]. Inspired by pangolins, an overlapping scaled design that allows users to concurrently realise on-demand thermal functionalities in tandem with shape-morphing and locomotion capabilities is introduced in this work. Heating of the metallic scales is controlled by an external 338 kHz RF field. The reported design allows for significant Joule heating ($\Delta T > 70\,°C$) at large distances (>5 cm), without compromising on the body compliance of these untethered soft robots (Fig. 1B). Design strategies to optimise the performance of the heating mechanism based on simulations and experiments are also presented. Enabled by our understanding of the system, we displayed advanced robotic functionalities, such as in situ demagnetisation and selective cargo release. Initial designs and results for untethered soft robots that can potentially perform clinically relevant tasks, such as mitigation of bleeding and hyperthermia, are also presented.

## Results

### Remote heating with alternating magnetic fields

Heating can be achieved using alternating magnetic or RF fields via two mechanisms: Joule heating or hysteresis losses[30]. In the former, when there is a change in the magnetic flux through a closed wire loop, an electromotive force (emf) is induced as described by Faraday's law. According to Lenz's law, the induced emf is such that the induced current in the loop produces a magnetic flux that seeks to oppose the change. This current, also known as an eddy current, flows through the conductor and heats the body up through Joule heating in the process. In the latter, magnetic hysteresis results in energy dissipation, which directly heats up the material[44]. This process dissipates significantly less heat as compared to Joule heating, contributing to less than 10% of total heating based on literature[30] and simulations conducted in COMSOL for steel ($3.88 \times 10^7$ W$m^{-3}$ and $1.42 \times 10^{-13}$ W$m^{-3}$ volumetric electric loss density and volumetric magnetic loss density, respectively).

To better illustrate the disparity in RF heating performance between Joule and magnetic hysteresis heating, we fabricated 1 cm × 1 cm square samples with a heating layer thickness of 100 μm, based on methods previously reported in literature[31,32,45]. These samples were then exposed to the same RF fields at a distance of 5 cm. From the results (Supplementary Fig. 1A), it was observed that there was no temperature increase for the methods relying on magnetic hysteresis (i.e., iron (II,III) oxide and mPDMS). Therefore, to generate the required temperatures, the devices had to be placed in the centre or on the surface of the RF coil, limiting the effective deployment range of these robots in meaningful biomedical applications. Only samples relying on Joule heating (i.e., 100 μm-thick aluminium and eutectic gallium-indium (eGaIn) surface) could record a temperature change at 5 cm. In particular, the 100 μm-thick aluminium sample was able to generate the largest temperature change at such distances. As demonstrated in this work, any changes in either the electrical conductivity or geometrical properties would lead to a significant deterioration in the heating performance. For a robot utilising liquid metals for remote heating, the robot's own deformation would also change the electrical conductivity and geometrical properties of the liquid metal layer and hence, affect the heating performance. As such, the use of liquid metals was ruled out in this work, as it would make the heating performance unpredictable and uncontrollable.

### Design for optimal Joule heating

The objective of the optimisation was to maximise Joule heating per scale while minimising the load carried by the soft robot, since the added weight of the metal scales for heating adversely affects actuation. Eq. 1 was defined to describe the underlying physics, direct the characterisation experiments, and hence the overall design strategy of these scales:

$$P_{in} = \rho V c_p \frac{\partial T}{\partial t} + H_L \qquad (1)$$

where $P_{in}$ is the input power, $\rho$ is the density of the material, $V$ is the volume of the material, $c_p$ is the specific heat capacity at constant pressure of the material, $\frac{\partial T}{\partial t}$ is the rate of change of temperature, and $H_L$ is the heat losses. Further details about Eq. 1 and the assumptions used in its construction are provided in *SI Appendix, Analysis of the system*. The heating performance is defined as the rise time ($\frac{\partial T}{\partial t}$) and the maximum temperature change in this work. From Eq. 1, we observed that the heating performance is dependent on the competing effects between heat generation, $P_{in}$, and heat loss, $H_L$. To maximise heating per scale, $P_{in}$ has to be maximised while minimising $H_L$ at the same time.

Based on an analytical analysis of Eq. 1, we observed that the electrical conductivity ($\sigma$) and geometrical properties, such as the length ($L$) and thickness ($w$), would be the dominant factors affecting the heating performance. Variations in $\sigma$, $L$ and $w$ directly affect the input power, $P_{in}$ – the induced current density through the material and also the magnetic flux passing through the scale changes when these parameters change. Moreover, variations in $L$ and $w$ directly affect the heating performance as they simultaneously affect the heat losses, $H_L$, and the rate of change of temperature, $\frac{\partial T}{\partial t}$ by changing the exposed surface area and the heating load, respectively.

As such, the characterisation experiments and simulations focused on studying the effects of these factors on the heating performance (Fig. 2A, B). The effects of the other material properties on heating performance are presented in *SI Appendix, Influence of material properties on heating performance*. Although RF heating on thin plates have been studied previously, the effects of this interplay between geometric and physical properties on heating efficiencies have not been studied extensively[46–49]. This serves as additional motivation for this section.

To maximise $P_{in}$, we first looked at the electrical conductivity. As the input power was directly proportional to the electrical conductivity from Eq. 1, a higher electrical conductivity should result in a higher temperature as the magnitude of the induced current density would be larger (Supplementary Fig. 2A). However, as the thickness of the plate is finite, the electrical conductivity cannot be increased indefinitely and there exists an optimal value (Fig. 2C, D). This optimal electrical conductivity decreases as the thickness of the samples increases. As the thickness of the scale increased from 50 μm to 250 μm, the optimal electrical conductivity for heating decreased from $1 \times 10^7$ S m$^{-1}$ to $2 \times 10^6$ S m$^{-1}$, respectively. For scales with a thickness of 100 μm, the optimal electrical conductivity was approximately $5 \times 10^6$ S m$^{-1}$, which was close to the electrical conductivity of tin. The simulations corresponded well with the experiments conducted with 100 μm thick scales, where tin ($\sigma = 8 \times 10^6$ S m$^{-1}$) produced the best heating performance (Fig. 2E). This maximum exists because the increase in the induced current density also leads to an increase in the magnetic field generated by per unit thickness of the scale to oppose the change in magnetic fields applied by the RF fields. Consequently, the magnetic flux from the RF coils penetrates less into the material (z-axis) for a scale with a larger electrical conductivity. Any additional material present above that thickness would only serve as a thermal load and not contribute to heating, resulting in the drop in the heating performance observed in Fig. 2D. For more details, please refer to *SI Appendix, Influence of electrical conductivity on maximum temperature*.

Apart from affecting the optimal thickness, changes in the electrical conductivity also resulted in a smaller skin depth, $\delta_{xy}$. The

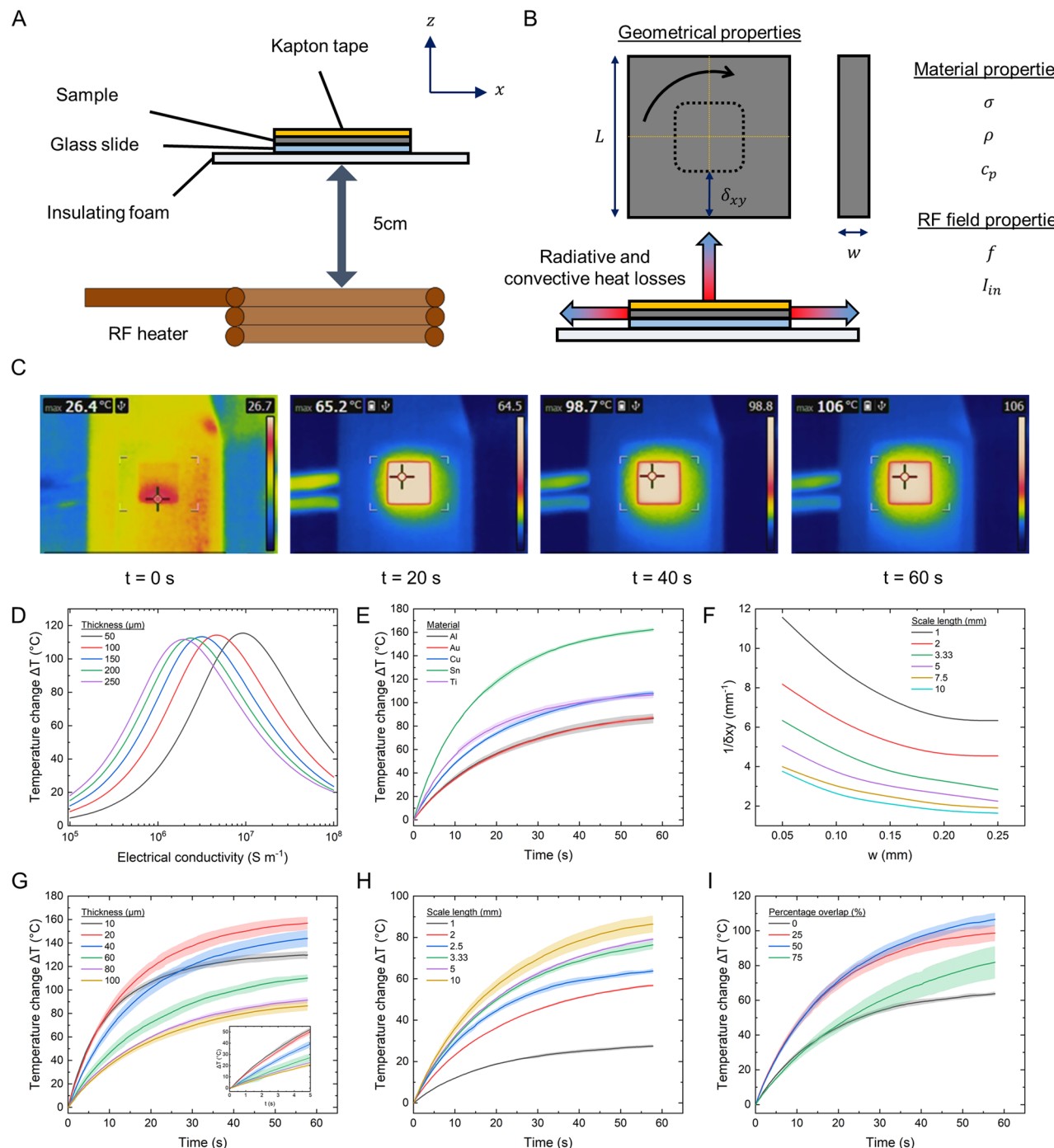

**Fig. 2 | Characterisation of heating performance. A** Schematic of experimental setup. **B** Identified parameters affecting Joule heating performance. **C** Infrared camera images tracking the temperature change of a 100 μm-thick aluminium scale over 60 s. **D** Simulated temperature at t = 60 s of the metal scales with varying electrical conductivities and thicknesses. **E** Temperature change of 100 μm-thick scales made from different materials over time (n = 6). Source data are provided as a Source Data file. **F** Simulated graphs for the selection of geometrical and material properties for optimal heating at different scale lengths. $\frac{1}{\delta}$ on the y-axis represents the inverse of the skin depth. **G** Temperature change of aluminium scales with different thicknesses over time (n = 6). Source data are provided as a Source Data file. **H** Temperature change of 100 μm-thick aluminium scales with identical areas but composed of different scale lengths over time (n = 6). Source data are provided as a Source Data file. **I** Temperature change of 100 μm-thick aluminium scales with different percentage overlap over time (n = 6). Source data are provided as a Source Data file.

formula for the calculation of the skin depth, $\delta_{xy}$, is defined by

$$\delta_{xy} = \sqrt{\frac{1}{\sigma\mu\pi f}} \qquad (2)$$

where $\sigma$ is the electrical conductivity of the material, $\mu$ is the permeability of the material, and $f$ is frequency of the applied RF field.

Since the currents were now confined to flow in a narrower region, the length of the scale had to be decreased correspondingly to ensure that the entire material plane still contributed to Joule heating. As such, we hypothesised that there was an optimal electrical conductivity for every thickness and length. A plot of $\frac{1}{\delta_{xy}}$ as a function of $w$ was generated to condense the identified effects (Fig. 2F). To validate this relationship, we characterised aluminium scales of different

thicknesses ranging from 10 to 100 μm. For aluminium, the value $\frac{1}{\delta xy}$ was calculated to be 7.1 mm⁻¹. Using the plots presented in Fig. 2F, the thickness of the scale had to be reduced to below 50 μm for optimal heating of a 10 mm scale. As shown in Fig. 2G, the heating performance increased as the thickness decreased. The best performance was recorded at 20 μm for aluminium. If the thickness was further reduced, as shown in the 10 μm thick aluminium scale, the maximum temperature generated started to drop. On top of generating a higher final temperature, another advantage of using thinner plates was that the rate of temperature rise was increased. As the plate thickness was decreased from 100 μm to 10 μm, the initial slope of the temperature-time graph increased (Fig. 2G, Supplementary Fig. 1H, I). For more details, please refer to *SI Appendix, Influence of electrical conductivity on temperature rise time.*

As the use of scales with hierarchical arrangement to enhance flexibility and mechanical compliance in biological systems has been well-studied[50,51], we looked into the effects of breaking up a single large structure into one composed of smaller scales. In this regard, the single 1 cm² metal scale was replaced with smaller scales of the same area, material, and thickness. As the length of the scales decreased, the heating performance decreased. The drop in the heating performance occurred because the magnetic flux passing through each scale decreased, resulting in a lower induced emf and hence, temperature (Supplementary Fig. 2D). Even though the total area was kept constant at 1 cm², the maximum temperature dropped dramatically, up to 75% as compared with the uncut piece – since the maximum amount of heat that could be generated per scale was now reduced (Fig. 2H and Supplementary Fig. 2E). This indicates that there is a fundamental trade-off between mechanical compliance and heating with the use of scales.

However, unlike other animals, such as armadillos[52], alligators[53], and lizards[54], which have scales arranged in a non-overlapping (i.e., osteoderm) configuration, the pangolin has overlapping scales, each of which are bonded directly to the underlying soft skin layer. This degree of overlap, or imbrication, ranges from 0.5 to 0.8 depending on the species[43]. We hypothesise that overlapping the scales would be advantageous for heating as it would increase the effective heating volume (i.e., $P_{in}$) while keeping any increase in surface area minimal. As an example, for an equivalent 50 μm-thick sample with a fixed 1 cm² area, there would be a 75% increase in volume but only a 5.7% increase in the exposed surface area at 50% overlap. Since $H_L$ is only proportional to the exposed surface area for a given change in temperature, this results in an overall gain in temperature since more volume is exposed to the RF field. As observed from Fig. 2I, increasing the number of plates on the robot by overlapping the scales increased the final temperature after 60 s of RF exposure, by up to 67% depending on the configuration used. Using this strategy, the 100 μm-thick scales with 50% overlap was able to perform as well as a 60 μm-thick unscaled 1 cm² sample. This demonstrated that the pangolin-inspired overlapping design was able to compensate for the decrease in the temperature arising from the division of a larger scale into smaller scales whilst still providing the necessary mechanical compliance. The heat produced by the scales is also highly reliable and repeatable, with the maximum temperature reached varying by less than 5% over 30 cycles. Even after it was autoclaved, the heating performance degraded by less than 5% (Supplementary Fig. 2G, H).

In summary, to achieve the best heating performance, the required mechanical compliance must be taken into consideration as it will determine the maximum size of a scale of length *L*, and hence the thickness of the scale used (Fig. 2F). Materials with a lower $\rho V c_p$ (i.e., lower mass, volume and/or a specific heat capacity) should be used to increase the rise time. Moreover, to overcome the disadvantages conferred by inducing cuts on the structure to increase the mechanical compliance, overlapping structures could be used. For ease of

reference, the effects of changing the parameters are summarised and provided in Supplementary Table 1.

## Mechanical deformation performance

After optimising the heating performance through the pangolin-inspired conductive scales, we verified the mechanical deformation performance of the overall soft robot to observe the effect of the scales on its locomotion. In this regard, the tests studied the response of the magnetic composite after addition of the metal scales. In the first series of experiments, the flexural compliance of the structure was examined by performing a three-point bending test on the hybrid robot in various configurations and percentage overlaps (Fig. 3A). In this work, intrados is defined as the configuration where the scales are facing up and extrados for the converse, where the scales are facing down. This would allow us to estimate the forces required to bend the composite structure and hence, the strength of the external magnetic fields required to actuate the robot.

First, the effects of the scale size were studied. In this regard, non-overlapping samples with different scale lengths were fabricated from a 50 μm-thick aluminium sheet. Once the scaled structures were introduced, the maximum stress experienced during the three-point bending test decreased by 87.9% from 8.3 MPa to 1 MPa (Fig. 3B). This brought the stress levels to an order that was comparable to the maximum stress experienced (0.18 MPa) by the magnetic polymer without any scales (mPDMS), highlighting the mechanical advantage of adding scales to the structure. As the length of the scales decreased from 5 to 1 mm, the maximum stress was further reduced to 0.68 MPa, a value only 3.7 times larger than mPDMS. This maximum stress was even lower for overlapping structures, dropping by an additional 85%, from 1.0 MPa to 0.15 MPa for the structures with 0% and 75% overlap, respectively (Fig. 3C). At 0.15 MPa, this meant that the structure was equally easy to deform as the mPDMS. Similar to how individual pangolin scales are directly connected to the soft skin layer and not interconnected[43], the scales on the robot were directly bonded to the magnetic polymer and were not bonded to each other. As such, the area directly bonded to the magnetic polymer decreased at increasing percentage overlaps. The scale, at increasing percentage overlaps, constrained less of the polymer and made the mechanical bending behaviour closer to that of the unscaled structures. In contrast, for the design with no overlap, the whole scale still rests on the surface of the soft magnetic polymer matrix. As such, the only way that the maximum stress could be reduced was to decrease the size of the scales.

As we were concerned about the load vs. deflection behaviour before and after addition of the scales, the flexural chord modulus of elasticity of the structure ($E_{FC}$) was calculated. A lower modulus of elasticity value was desired as it would mean that a lower force had to be applied to the structure per unit of deformation. As the length of the scales decreased from 5 to 1 mm, the modulus of elasticity dropped from 31.7 to 19.5 MPa. For comparison, the values for mPDMS and the structure without any scales were 9.3 and 583.7 MPa, respectively. As such, this meant that increasing the externally applied fields was a viable method to actuate these robots – it required at most three times more force per unit of deformation. We also characterised the effects of using different materials or thicknesses on $E_{FC}$. In this regard, we observed that $E_{FC}$ was predominantly a geometrical property. Unscaled samples made from other materials differed at least one or two orders of magnitude in $E_{FC}$ as compared to mPDMS. However, this value dropped to the same order of magnitude once the scales were added. The same trends were also observed even when the thickness of the scales were changed (Fig. 3D).

Finally, we looked at the flexural compliance of the overlapping scaled robot in various configurations and orientations. Apart from the extrados test configuration, similar trends were also observed for the other test configurations (i.e., significant drop in maximum stress and $E_{FC}$). Moreover, $E_{FC}$ of the overlapping scaled robot in a transverse

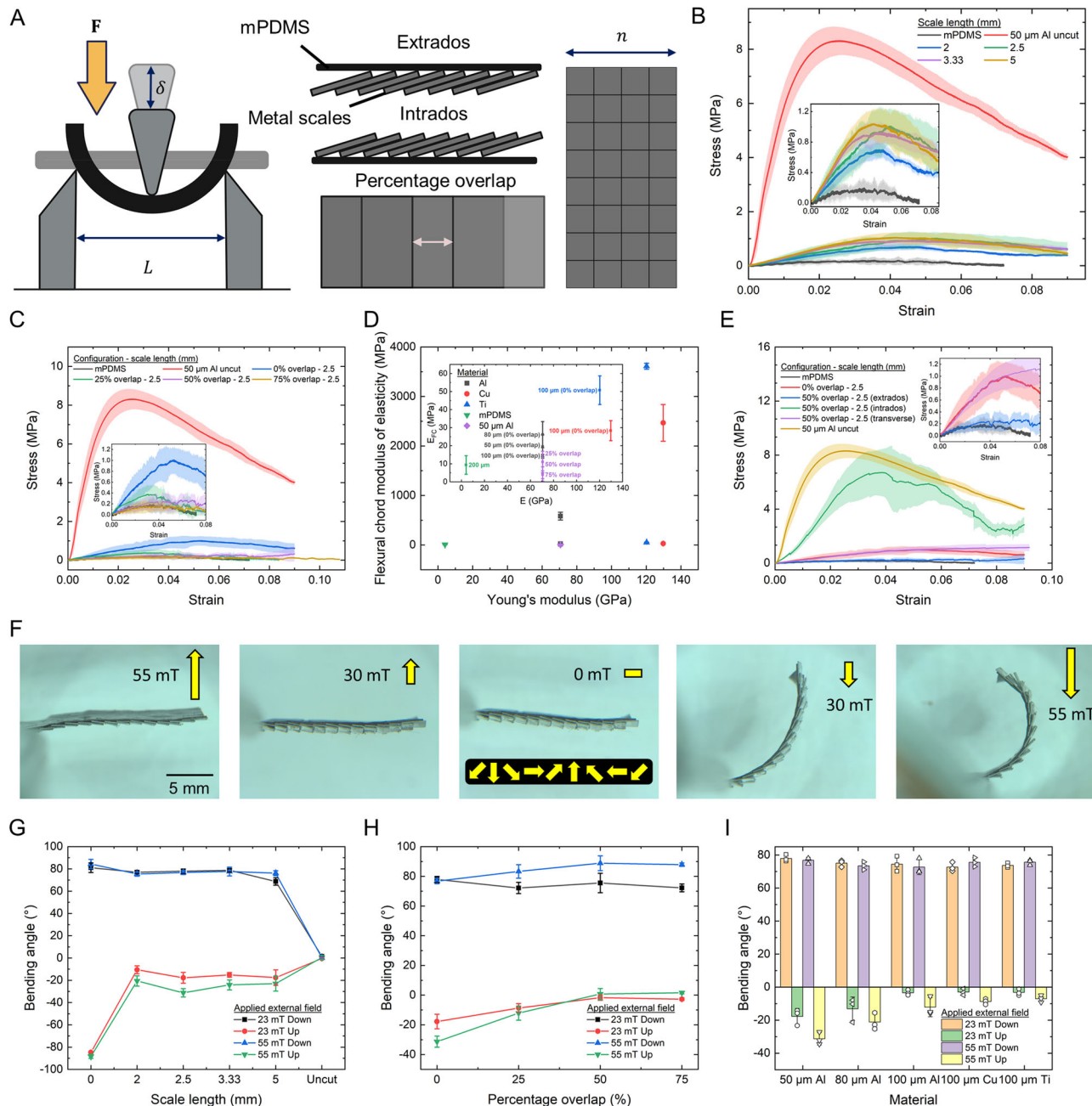

**Fig. 3 | Characterisation of mechanical performance. A** Schematic of experimental setup. Figure created with biorender.com. **B** Stress-strain curve for a 20 × 10 × 0.25 mm sample with different scale lengths (n = 3). Source data are provided as a Source Data file. **C** Stress-strain curve for a 20 × 10 × 0.2 mm sample bonded to 0.05 mm aluminium with different percentage overlap (n = 3). Source data are provided as a Source Data file. **D** Comparison of the flexural chord modulus of elasticity ($E_{FC}$) for different materials and thicknesses (n = 3). Source data are provided as a Source Data file. **E** Stress-strain curve for a 20 × 10 × 0.2 mm sample bonded to 0.05 mm aluminium at 50% overlap with different test configurations

(n = 3). Source data are provided as a Source Data file. **F** Deflection of a 20 × 10 × 0.2 mm sample bonded to 0.1 mm Al at 50% overlap at different applied external magnetic fields. Inset shows the magnetisation profile of the robot. **G** Deflection angles for a 20 × 10 × 0.25 mm sample with different scale lengths (n = 3). Source data are provided as a Source Data file. **H** Deflection angles for a 20 × 10 × 0.2 mm sample bonded to 0.05 mm aluminium with different percentage overlap (n = 3). Source data are provided as a Source Data file. **I** Comparison of the deflection angles for different materials and thicknesses. Source data are provided as a Source Data file.

configuration, as denoted by the gradient of the stress-strain graphs, was comparable to the value of the scaled robot with no overlap tested in a longitudinal configuration (Fig. 3E). Regardless of the overlapping scales in the longitudinal direction, the robot had a non-overlapping configuration in the transverse direction which gave rise to a similar $E_{FC}$. We also note that the intrados configuration had similar levels of $E_{FC}$ as compared to the uncut scale (246.5 vs. 583.7 MPa). Although not exploited in this work, such a capability could potentially allow the

robot to lift or support large loads, whilst still remaining deformable enough for magnetic actuation-based shape programming.

The second set of characterisation experiments focused on the bending response of the magnetic composite after addition of the scales. In this regard, the angle between the tip and the fixed support when subjected to different external magnetic fields was compared. Anti-clockwise angles were defined as positive in this work (Fig. 3F). Similar to the flexural tests performed earlier, the tests focused on the

effects of the scale length, overlap and material selection on the bending angles. First, we looked at the bending angles of the robot after the scales have been introduced. Similar to the experiments for flexural compliance, the same trends and observations for the scaled and overlapping samples were obtained. In this regard, it was observed that once the scales were added, the robot was able to achieve similar levels of deflection as the mPDMS (Fig. 3G). Although there was a slight decrease in the anti-clockwise deformation (81° vs 76°), this deformation angle was noted to be insensitive to the strength of the externally applied magnetic field tested (23 and 55 mT). However, the robot's deflection in the direction of the scales was compromised and the robot was unable to reach the original levels of deflection. Deflection in the direction of the scales would cause the plates to come into contact with each other and as such, limit the maximum deflection. The same trends were also observed in the overlapping samples (Fig. 3H). Finally, we looked at the deflection angles with respect to different materials and found that the deflection angles are predominantly a geometric property and are independent of the material properties. When the thickness of the scales were reduced, an increase in the clockwise deflection was noted. For thinner scales, the magnetic PDMS would be free to deflect more before the scales come into each other (Fig. 3I). The experimental results were also consistent with the simulation results obtained in COMSOL (Supplementary Fig. 3).

In summary, we found that the pangolin-inspired overlap structures offered the best mechanical compliance, even matching the mPDMS at 75% overlap. Similar to the animals found in nature, we find that a higher degree of overlap offers better mechanical compliance. Scales found on African tree pangolins, have a higher overlapping ratio and as such, are more flexible and can perform a larger range of motion which might be necessary for survival in their habitat. In contrast, Chinese pangolins have a lower overlapping ratio and mainly live in burrows on the ground[43,55,56]. This means that the overlapping design could potentially allow a larger scale for better heating to be used without compromising the mechanical compliance.

## Multifunctional robotic capabilities

Finally, we demonstrate the capability of such robots based on active locomotion with magnetic actuation coupled with on-demand functionalities enabled by the addition of the heating scales. The functionalities shown in this work can be broken down into two main categories. The heat could be used to change certain properties of the robot on demand and then harnessed to achieve new functions, such as selective cargo release and in situ demagnetisation. Alternatively, the heat produced could potentially be directly used by untethered robots to perform medical treatments involving heat, such as hyperthermia for cancer treatment or to mitigate bleeding in hard-to-reach regions.

Aluminium was selected as the material for the subsequent demonstrations because it had a high electrical conductivity, allowing the thickness of the scale to be decreased. This, coupled with the low density and the non-ferromagnetic nature of aluminium, meant that it would be easier to actuate the robot and would also eliminate any possibility of the material interfering with magnetic actuation. The pull-out force of the aluminium scale on mPDMS was experimentally determined to be approximately 800 mN, implying that the scales are unlikely to fall off during actuation (Supplementary Fig. 4). With this, we demonstrated how an untethered robot comprising 100 μm aluminium scales with 50% overlap was able to actuate and move in a stomach phantom with a 65 mT rotating magnetic field even with a non-optimal configuration for heating and locomotion (Supplementary Movie 1).

In the first category, we demonstrate a version of an untethered magnetic robot that can perform on-demand in situ demagnetisation, a function enabled by the heating scales. In this demo, a robot with non-overlapping 50 μm aluminium scales was used to enable in situ

demagnetisation, thereby allowing for the magnetisation profile on the robot to be changed in situ. Non-overlapping scales were used to ensure even heating of the magnetic polymer. In the first phase, the robot employed a rolling strategy for locomotion because it had a pre-programmed sinusoidal magnetisation profile imparted to it prior to deployment. After reaching the target location, the robot was exposed to the RF field. In doing so, the scales and the magnetic polymer were heated to temperatures above 219 °C. As this was above the Curie temperature of the magnetic particles (MQFP-10-8.5HD-20180, Magnequench) embedded in the soft matrix, the particles shifted to a paramagnetic state and the magnetisation profile previously encoded was lost. After in situ demagnetisation, the robot was observed to be unresponsive to any input magnetic fields, indicating that the robot no longer had any magnetisation profiles (Fig. 4A and Supplementary Movie 2). This was also confirmed separately when the magnetic flux density on the robot was measured (Supplementary Fig. 5). Only after subsequent magnetisation with a uniform 1.8 T external magnetic field was the robot able to locomote again. The temperature required and magnetisation field could be lowered even further if magnetic particles with a lower coercivity and Curie temperature are used. Due to the difference in the magnetisation profile, the robot now utilised tumbling rather than rolling for locomotion (Fig. 4B). Such system could enable an in situ change of the robot's magnetisation profile, which could enhance the functionalities of these robots. Specifically, miniature magnetic soft robots reported in literature to date have a fixed magnetisation profile which, after deployment, cannot be changed in situ. As such, the response of the robot to an external magnetic field cannot be changed after it is deployed. Although reprogrammability by heating above the Curie temperature had been demonstrated previously[57], the process was not in situ and a laser was used to locally heat up the polymer. This process allowed very precise programming of the magnetisation profile, yet had limited use in biomedical applications inside the human body as the laser required line of sight for heating. As such, this demonstration highlights the ability of this method to allow these untethered robots to potentially change its magnetisation profile on the fly. This further opens up the design space and enhances the performance of untethered miniature robots, since a single robot design can now potentially change and adapt its locomotion based on the environment accordingly in situ. Moreover, as demonstrated in Supplementary Movie 2, the demagnetised robot is unresponsive to the actuating fields. As such, through judicious placement of the heating plates, demagnetisation of those parts could enable selective actuation of the robot for specific tasks after the robot has navigated to the target location. Alternatively, this demagnetisation feature could allow for better control of robotic swarms. For instance, untethered robots can be deployed singularly and then deactivated, thereby allowing another robot to be deployed to the same region without the concern that the actuating fields for the second robot would unintentionally actuate the first as well.

Next, we demonstrate how the heating scales can enable selective on-demand cargo release. Magnetic miniature robots typically operate at a distance from the magnetic actuation coils. This, combined with their small size, imply that the robots perceive the external magnetic field as a homogenous field. As a result, a single robot can only have a monotonic response to the externally applied magnetic field. In this variant of the robot, we exploited the different heating rates of the scales to enable selective cargo release. The cargo was secured to the robot with beeswax, which had a melting point of 61 to 65 °C. A 50 μm and 80 μm aluminium scale was placed under the blue and green cargo, respectively, to heat up the beeswax (Fig. 5A). When exposed to the RF field, the 80 μm-thick aluminium scale was able to reach the targeted temperature approximately 1 s faster than the 50 μm-thick aluminium scale (Supplementary Fig. 6). This allowed the robot to only release the blue cargo but not the green cargo (Fig. 5B and Supplementary Movie 3). It is important to note that the temperature

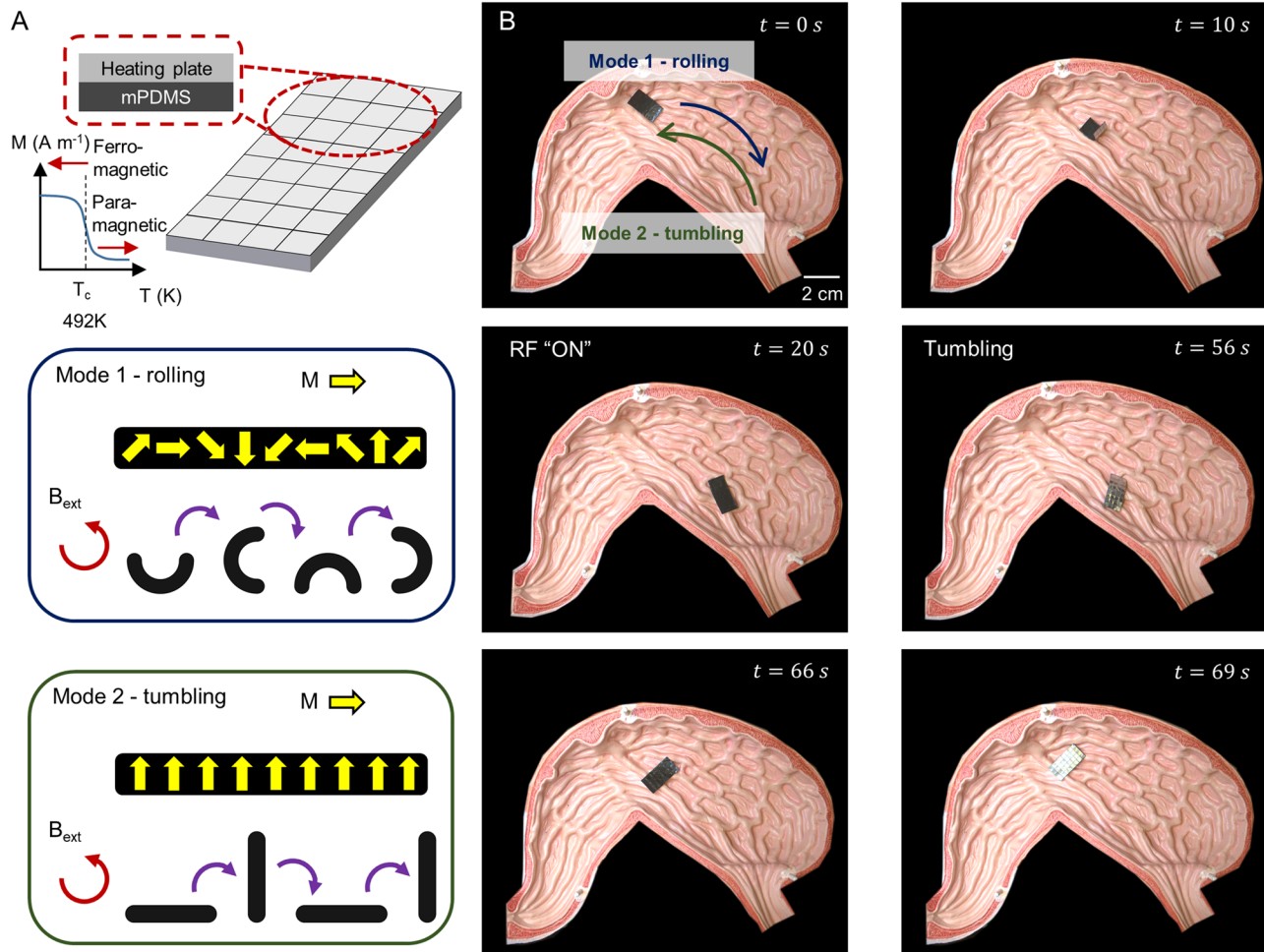

**Fig. 4 | Enhanced functionalities of untethered miniature robots. A** Schematic of the untethered magnetic robot which can perform in situ demagnetisation to switch the locomotion modes from Mode 1 – rolling to Mode 2 – tumbling. Inset shows the magnetisation profile and the response of the robot to an externally applied magnetic field. Figure created with biorender.com. **B** Deployment of the robot in a stomach phantom.

required was slightly higher than the melting point of beeswax as the addition of beeswax increased the thermal load. Such a feature could not be easily integrated with other heating methods, including those involving Joule heating (i.e., liquid metal droplets mixed in the polymer matrix) because the heating occurred homogenously throughout the liquid metal droplets dispersed inside the robot. The main limitation of this robot design lies in the selection of adhesive to secure the cargo to the robot. Although the beeswax used in this demonstration is biocompatible and can be safely ingested, the beeswax re-solidifies upon cooling. This does not pose a problem in this work because we intend to deploy the robot in the GI tract, where the wax can be safely excreted by the human body. However, to further extend the clinical utility of these robots, studies should be conducted to identify adhesives, which thermally degrade upon heating into compounds that can be safely excreted or absorbed by the body. On a related note, any functionality requiring heat activation could also be implemented. For instance, with the addition of thermal adhesives on the heating scales, we were also able to show that the magnetic robot was able to adhere to P100-grade sandpaper which has an average particle diameter of 162 μm (Supplementary Movie 4).

### Untethered heating robots towards medical applications

In the latter category, we demonstrate how the heat could be used directly to perform medical treatments in the gastrointestinal (GI) tract. First, we showed how a robot with 50 μm-thick aluminium scales

with 50% overlap could be easily incorporated into a standard size "0" gelatine capsule for oral deployment (Fig. 6A). Using such a method allowes us to non-invasively deploy these robots in tortuous hard-to-reach regions, such as the small intestine, which is difficult to access using established techniques, such as endoscopy or colonoscopy. Moreover, to account for the highly heterogeneous environments inside the body, simulations were performed to determine the heating performance of the scales across different distances and convective heat transfer coefficients. This would provide insights on the heating performance of the robot under more realistic conditions. Based on the additional simulations performed (Supplementary Fig. 7), we observed that for a similar decrease in heating performance (i.e. final temperature of the metal scale after 60 s of RF exposure), the distance from the coil had to be increased by 3 times while the convective heat transfer coefficient had to be increased by 1000 times for a given RF input. This implies that the heating performance is more sensitive to changes in the magnetic flux (i.e., distance) as opposed to changes in heat losses (i.e., environmental conditions).

To demonstrate the potential clinical utility of such a robot, we simulated bleeding inside an ex vivo porcine stomach. The bleeding rate was set at $1\,\mu L\,s^{-1}$, to mimic capillary flow rates in the GI mucosa[58]. The robot was able to navigate to the bleeding site and upon application of a 3 s RF pulse, stop the bleeding at the site (Supplementary Movie 5 and Fig. 6B). Although this demonstration was conducted in stomach, this robot could potentially be deployed in other parts of the

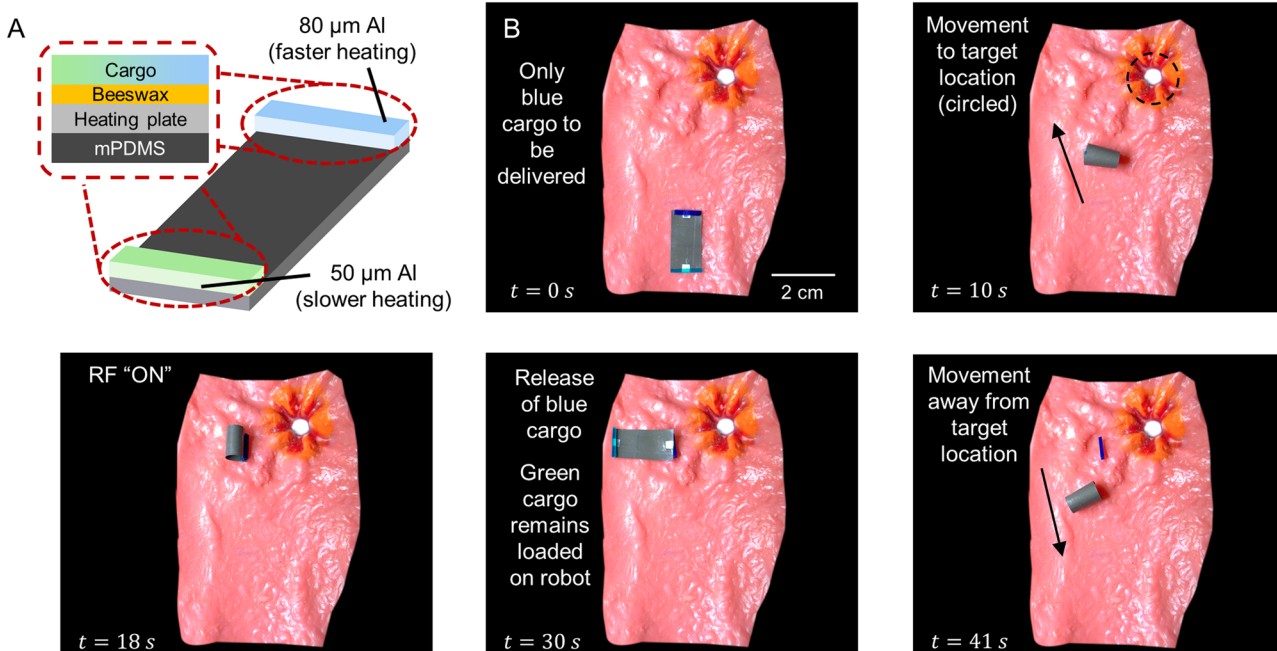

**Fig. 5 | Enhanced functionalities of untethered miniature robots. A** Schematic of the untethered magnetic robot which can perform selective cargo release. The selective cargo release is enabled by exploiting the different heating rates of aluminium of different thicknesses. It can also be enabled by using different materials. Figure created with biorender.com. **B** Deployment of the robot in a tissue phantom.

GI tract as well. This robot design could be an enabling technology for the treatment of GI bleeding in hard-to-reach sites. One such example would be bleeding in the small intestine, which accounts for almost 5% of all GI bleeding, but remains challenging to diagnose and treat with conventional treatment methods[59]. Such an untethered robot could also offer clinicians with a less invasive alternative for elective procedures for small intestinal bleeding with various etiologies (i.e., ulcers, angiodysplasias, polyps or tumours) especially when coupled with medical imaging feedback. For this reason, we also explored how the robot is compatible with existing medical imaging modalities, such as ultrasound imaging (Supplementary Movie 6 and Fig. 7A). Such a capability could also enable novel non-invasive treatments, such as hyperthermia for the treatment of cancer[60]. In this regard, we subjected tumour spheroids in direct contact with the heating scales to the RF field and observed that the tumour spheroids were destroyed after just 5 min of heating at 60 °C (Fig. 7B, C). Results from the initial biocompatibility tests for the long-term use of aluminium inside the human body and how locomotion will be affected by the environment are presented in *SI Appendix, Initial Biocompatibility Studies* and *SI Appendix, Environmental impact on locomotion*, respectively. Moreover, the safety of deploying these robots inside the human body can be further enhanced by adding fillets to the scales to reduce the likelihood of tissue puncture, without adversely affecting the overall heating performance (Supplementary Fig. 11).

## Discussion

In this work, we introduced a design to enable heating over long distances for untethered miniature robots. Inspired by pangolins found in nature, the scaled design introduced in this work allowed for two different competing requirements, namely the compliance and heating performance, to be concurrently realised on a single untethered robot. Although hierarchical structures have been implemented in literature primarily for their mechanical properties, we find that implementing the overlapping structures found in pangolins is also advantageous to remote heating. This allowed us to achieve significant heating on demand ($\Delta T > 70$ °C) at large distances (> 5 cm) within a short period of

time (<30 s) without sacrificing on the bending compliance of the robot. Guidelines to optimise the heating performance and bending compliance were established. Most importantly, the mechanism was small and light enough to be mounted on existing magnetic soft millirobots. Enabled by the remote heating capabilities of the design in tandem with locomotion, integrated robots displaying advanced robotic functionalities such as in situ demagnetisation and selective cargo release were demonstrated. Initial steps for untethered robots targeting potential biomedical applications, such as mitigation of bleeding and hyperthermia were also demonstrated. These functionalities highlight the possibility of using such untethered robots in medical applications.

For successful translation, future studies should concentrate on four technical aspects on top of application-specific issues, such as how residual intestinal content might still be present after flushing of the GI tract and how it may interfere with locomotion. Firstly, since both actuation and remote heating involve magnetic fields, future work should look into the possibility of creating a magnetic system in which both the high and low-frequency fields can be given with a single magnetic actuation setup. Doing so would significantly reduce the time and effort required to operate such a robot, while allowing for more accurate control of the magnetic fields. This would then allow true in situ reprogramming of the magnetisation profile to be achieved (i.e., both demagnetisation and magnetisation). Secondly, the heating efficiencies of RF fields could potentially be further increased by exploiting secondary effects, such as the proximity effect or through the addition of insulating layers such as Parylene C on the sides of the scales. This would generate more heat and further extend the operating range of the robot. Thirdly, regarding the safety of biological tissues to such an RF exposure, a maximum current of 621.6 A at 338 kHz was applied by the RF heater positioned 5 cm away for up to 15 min in all ex vivo demonstrations. This translates to a magnetic field intensity of 34.6 kA m$^{-1}$ and is comparable to the levels of RF that is being applied in in vivo experiments or clinical trials currently (Supplementary Table 2). Although this highlights the potential of using such devices in a clinical context, in vivo tests should still be conducted

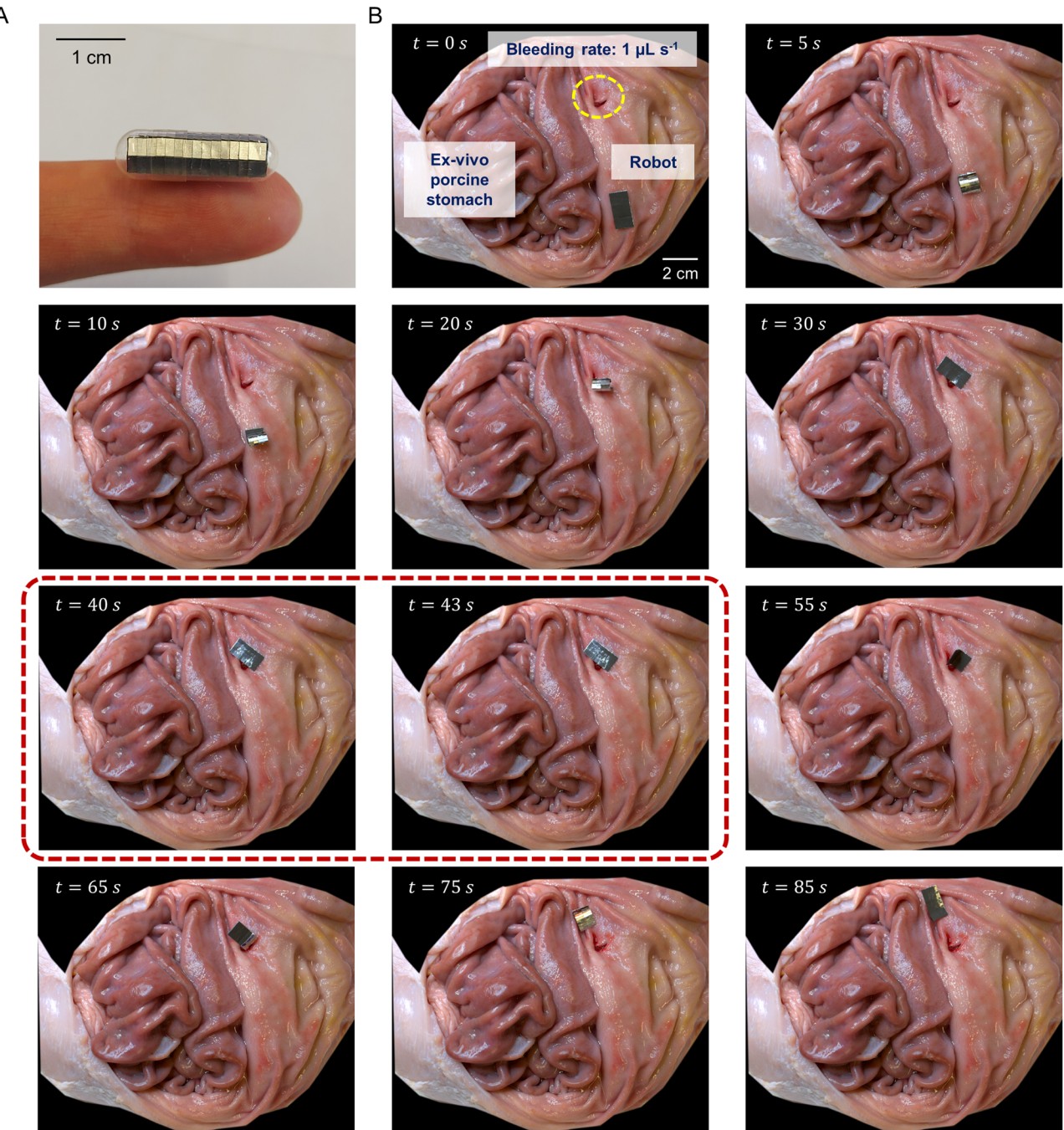

**Fig. 6 | Ex vivo demonstration directly utilising heat energy to mitigate blood loss. A** A 20 × 10 × 0.2 mm robot with 50 μm aluminium scales at 50% overlap inside a standard size "0" gelatine capsule (21.2 × 7.3 mm). **B** The integrated robot moves to the target location inside an ex vivo porcine stomach. Upon application of a 3 s RF pulse (frames circled in red), no more bleeding at the site was observed.

to definitively conclude the safety of such an exposure and also the upper limit of exposure. A higher current would generate more heat for other applications, further extend the operating range of the robot or reduce the duration of RF exposure. Lastly, the use of better materials in terms of biocompatibility and biodegradability should be investigated. An example would be Dermabond, which is currently used in clinics for topical wound closures and hence, could potentially provide the necessary adhesion of the scales to mPDMS whilst simultaneously offering better biocompatibility. Further studies will have to be conducted to assess the suitability of this adhesive for applications inside the body as the working environment inside the human body varies greatly from the outside. Biocompatible and biodegradable

materials such as FePt and hydrogels as substitutes for mPDMS should also be considered. Addressing these issues would further enhance the capabilities of the untethered robot and can potentially unlock a new suite of minimally invasive long-term medical procedures, which are currently unavailable.

## Methods

### Cell culture

Human skin fibroblast cells, BJ (CRL-2522, American Type Culture Collection), were cultured in a humidified 37 °C and 5% $CO_2$ atmosphere using 75 cm² polystyrene cell culture flasks in high-glucose DMEM supplemented with 10% foetal bovine serum (FBS, Gibco) and

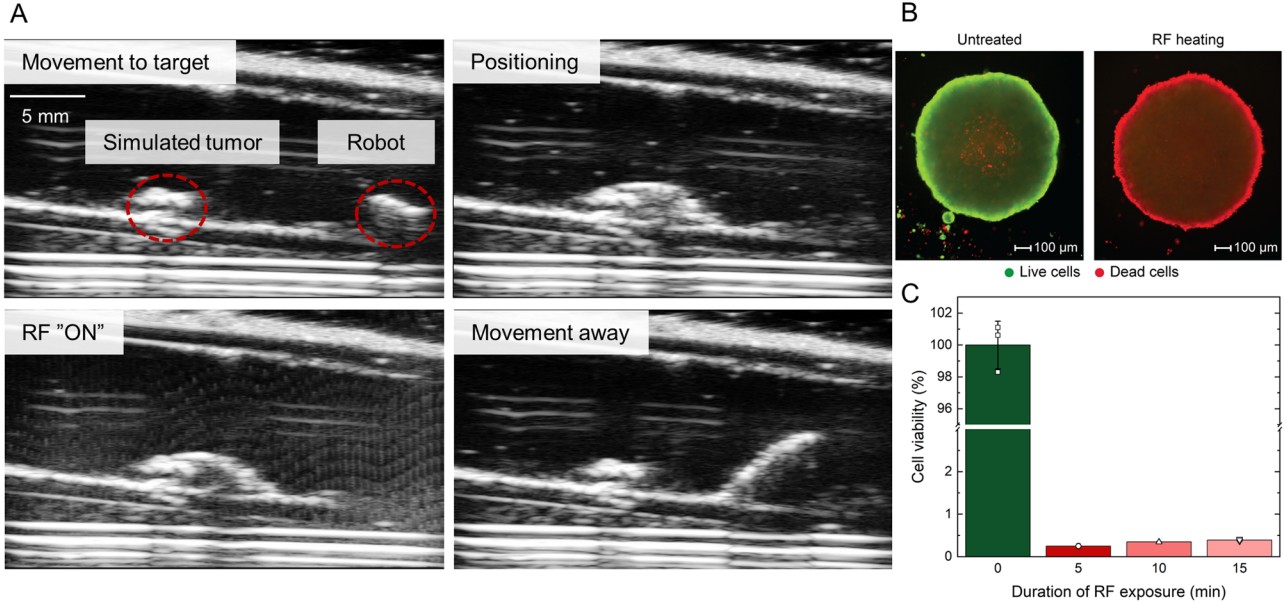

**Fig. 7 | Ex vivo demonstration directly utilising heat energy for hyperthermia. A** Ultrasound guided robot operating inside an ex vivo porcine small intestine with a simulated tumour. Small intestine is filled with DI water. (**B**) Representative fluorescence images of RF exposed HT-29 tumour spheroids stained with calcein-AM/ethidium homodimer−1 after 24 h of incubation. **C** Viability of HT-29 tumour spheroids after different durations of RF exposure (n = 3). Three spheroids were independently tested once for the experiments. Source data are provided as a Source Data file.

1% penicillin/streptomycin (Gibco). Cells with passage numbers less than 10 were used, and surface detachment was performed using trypsin (0.25 wt/wt%)/EDTA solution (Gibco) when they reached 80% confluence.

Human colorectal adenocarcinoma cells (HT-29, American Type Culture Collection), were cultured in a humidified 37 °C and 5% $CO_2$ atmosphere using 75 cm$^2$ polystyrene cell culture flasks in high-glucose DMEM supplemented with 10% foetal bovine serum (FBS, Gibco) and 1% penicillin/streptomycin (Gibco). Cells with passage numbers less than 10 were used, and surface detachment was performed using trypsin (0.25 wt/wt%)/EDTA solution (Gibco) when they reached 80% confluence.

### Biocompatibility assay

Biocompatibility assays were performed on (i) 1 cm × 1 cm aluminium squares with a thickness of 100 μm and (ii) different concentrations of aluminium powder. BJ fibroblasts, CRL-2522, were seeded on the top of 1 cm x 1 cm aluminium substrates and the viability of cells after 72 h of incubation was analysed. After 72 h of incubation, the fibroblasts were stained with a Live/Dead Cell Imaging Kit (Invitrogen) according to the supplier's instructions. Images of living (Ex/Em: 488/520 nm) and dead (Ex/Em: 528/617 nm) cells were obtained using a fluorescent microscope (Nikon Eclipse Ti-E).

The human skin fibroblast cells, CRL-2522, were seeded in a black/clear bottom 96-well plate (Corning) at a concentration of 1 × 10$^4$ cells per well the day before the experiment to allow for the attachment of cells. After 24 h of incubation, the cells were treated with different concentrations of aluminium powder (37.5 μg mL$^{-1}$, 75 μg mL$^{-1}$, 150 μg mL$^{-1}$, 300 μg mL$^{-1}$, 600 μg mL$^{-1}$ and 1200 μg mL$^{-1}$ in DMEM) in triplicates for 24 h and 72 h. After incubating for the given periods, the cellular viabilities of BJ fibroblasts were measured using the CellTiter-Glo assay (Promega)[61]. The luminescence values were measured in an opaque 96-well plate using a plate reader (BioTek's Synergy 2, Winooski, VT, USA). The viabilities of treated fibroblasts with different concentrations of aluminium powder were expressed as a percentage of the untreated fibroblasts, which was taken to be 100%.

To test the stability of aluminium in various fluids in the GI tract, 1 cm$^2$ aluminium scale of 100 μm thickness was submerged in 2 mL of simulated gastric fluid (3082.1000, Th. Geyer), intestinal fluid (D020-00, Th. Geyer) and high-glucose Dulbecco's Modified Eagle Medium (DMEM, Gibco). The absorbance of the solutions was measured with a plate reader (Infinite M Plex, Tecan) at 300 nm.

### Characterisation of heating scales

Unless otherwise stated, square samples with a side of 1 cm were used for the characterisation experiments. These samples were laser cut from a larger foil with the appropriate thickness. The samples were cleaned with IPA after laser cutting and bonded to a 12 × 12 mm microscope cover glass slide (0101000, Paul Marienfeld GmbH) with a 50 μm layer of silicone adhesive (Sil-Poxy, Smooth-On Inc.) and left on a hot plate to cure at 90 °C for 5 min. A 50 μm Kapton layer (KAP22-075, Thorlabs) with identical dimensions to the sample was attached to the top surface of the sample to eliminate any discrepancies arising from different emissivities when imaged with an infrared camera. The RF heater was turned on for 60 s at 621.6 A. During the characterisation experiments, all the samples were placed 5 cm away from the edge of the RF coil. The samples were insulated at the bottom to prevent heat loss by conduction. The RF heater was turned on for 60 s and the resultant temperature rise was tracked with an infrared camera (ETS320, Teledyne FLIR) placed 7 cm above the samples (Fig. 2C). For the autoclave repeatability tests, the heating scales were autoclaved (VX-150, Systec) at 121 °C for 20 min followed by 10 min of dry time, with a final temperature of 120 °C. Three samples were tested twice each. The data from the infrared camera was saved on a computer using the software FLIR Thermal Studio. After the experiments, the data was exported from FLIR Thermal Studio. The average and standard deviation values across different runs were computed using MATLAB R2019A and visualised with Origin 2019b.

### Characterisation of mechanical properties

The test method was adapted from the D7264 ASTM (American Society for Testing and Materials) standard. In this regard, a 20 mm × 10 mm × 0.2 mm sample was tested on a micro 3 point bending test fixture (2810-411, Instron) of an Instron machine (5942, Instron) with the 5 or 10 N load cell, depending on the maximum load. The supporting pins were positioned 10 mm apart and the samples were

loaded at a rate of 20 µm s⁻¹. Stresses and strains were computed based on methods proposed in the standard. For the overlapping structures, the average height was used. The average height was defined as the height of the centre of the scale (Supplementary Fig. 12). Three samples were tested once each. The flexural chord modulus was calculated from the linear region of the stress-strain graph. This value ranged between 0.003 and 0.01 strain depending on the samples. The reported values are the engineering stresses and strains.

To characterise the deformation of the magnetic polymer with scales, the robot was placed in a uniform magnetic field generated by a Halbach array. The robot was fixed on one end by clamping with a pair of flat tweezers while the other end of the robot was free to deflect. The angle between the tip and the fixed support was analysed in ImageJ. Three samples were tested once each.

A pair of tweezers was clamped to the end of the Instron machine with a 5 N load cell while the scaled robot was clamped on the other to measure the pull-out force of the aluminium scale from PDMS. Tweezers were used to grip the scale by reducing the contact area. The grippers were then translated at a rate of 1 mm min⁻¹. The pull-out force was defined as the highest recorded force. Three samples were tested twice each. The scale to be pulled out was selected at random. The average and standard deviation values across different runs were computed using MATLAB R2019A and visualised with Origin 2019b.

### Demonstration of robotic capabilities

In all demonstrations, including those involving ex vivo tissues, the magnetic soft robot used had a standard size of 20 mm × 10 mm × 0.2 mm, unless otherwise stated. The RF coil was placed 3 cm away and operated at 338 kHz with an input current of 621.6 A. A cylindrical magnet (70 mm diameter, 35 mm height) was used to generate the magnetic fields required for remote actuation. The desired temperatures on the robot in the demonstrations were achieved by performing a calibration before the actual experiment. In this regard, the robot to be used in the demonstration was placed on a glass slide and exposed to the RF field while the temperature was tracked with a forward-looking infrared (FLIR) camera. The time taken to reach a temperature 10% higher than the desired temperature, to compensate for additional thermal losses, was then recorded. This time was then used during the actual demonstrations.

The robot for the thermal adhesive demonstration had a single continuous 2.5 mm × 10 mm × 50 µm aluminium heating scale which was bonded to a thermal bonding film (583, 3 M) of the same size. To bond the thermal film to the heating scale, a thin layer of 10:1 PDMS was applied and the resultant structure was left to cure at room temperature for 24 h. For the selective cargo release demo, the cargo was secured to the robot by dipping in liquid beeswax (243221, Sigma-Aldrich) and placing on the heating scales immediately.

### Demonstrations with ex vivo tissues

Pig organs and blood from a slaughterhouse were obtained, stored at 5 °C, and tested within 24 h upon receiving. The stomach and small intestine were used in this work. All organs were used as received without any additional cleaning steps. For the bleeding experiments, fresh blood was pumped into a cut induced by a scalpel with an external syringe pump at a rate of 1 µL s⁻¹. The robot was actuated with a handheld magnet. In the ultrasound-guided demonstration, the robot was positioned using a handheld magnet. A 20 mm × 10 mm × 0.2 mm robot with 50 µm aluminium scales at 50% overlap and 7.5 mm × 2.5 mm × 0.2 mm robot with 50 µm aluminium scales at 0% overlap were used in the bleeding and ultrasound demonstrations, respectively. The simulated tumour in the small intestine was created with 10:1 PDMS.

### Finite element analysis

Simulation results for deflection angles were generated with a custom-generated environment in COMSOL 6.0[57]. Simulation results for RF heating presented were generated with the commercially available finite element software, COMSOL 6.0. The simulations were conducted in COMSOL using the Magnetic Fields and Heat Transfer in Solids modules. The RF heater was modelled as a single conductor coil as per the dimensions and values provided by the manufacturer. Heat losses implemented along the boundaries were external natural convection and radiative heat transfer. The relevant material properties provided by COMSOL for the simulation (i.e. mechanical and electrical) were used without further modification. Simulation results for the thin films presented in Supplementary Fig. 1I were obtained by using the Magnetic Fields and Heat Transfer in Shells modules. Specifically, the plate was modelled as a two-dimensional (2D) plate and made 3D by approximating it as a non-layered shell of correct thickness.

### Sample preparation of samples for comparison of heating performance

All thicknesses of the heating component were fixed at 100 µm for a 1 cm² square sample. For the mPDMS and eGaIn sample, the liquid metal eGaIn (495425, Sigma-Aldrich) and neodymium-iron-boron (NdFeB) magnetic microparticles (MMPs, MQP-15–7, Magnequench) were mixed with PDMS (Sylgard 184, Dow Corning), at a weight ratio of 11:110:10:1 (MMPs: eGaIN: PDMS base agent: PDMS curing agent). The mixture was hand-stirred until a consistent emulsion was formed. This was determined visually and typically took 5 min of continuous stirring. The mixture was degassed for 30 min to remove the trapped air and poured into a mould. Excess mixture was then removed with a doctor blade. The mould was placed on a hot plate at 90 °C for 2 h to allow the composite to cure. For the mPDMS and eGaIn surface sample, the 100 µm-thick liquid metal eGaIn layer was added to the surface of a 11:10:1 (MMP: PDMS base agent: PDMS curing agent) substrate by doctor blading.

mPDMS samples were fabricated by mixing the constituents at a weight ratio of 11:10:1 (MMPs: PDMS base agent: PDMS curing agent). Similarly, the iron (II,III) oxide (637106, Sigma-Aldrich) samples were fabricated with a weight ratio of 11:11:10:1 (MMPs: iron (II, III) oxide, PDMS base agent: PDMS curing agent). After mixing and degassing the mixture for 30 min to remove the trapped air, excess mixture was removed with a doctor blade, and the mould was left on a hot plate at 90 °C for 2 h to allow the composite to cure.

### Scaled soft robot fabrication

To fabricate the device, grooves corresponding to the final design were laser cut into a metal sheet of the desired thickness. The grooves did not cut through the entire depth of the metal sheet to facilitate subsequent fabrication processes. For the overlapping structures, an additional step of engraving the metal at a lower power to indicate the extent of overlap was performed. The degree of overlap is defined as the exposed length over the total length of the scale in this work. The top and bottom surfaces were then cleaned with isopropyl alcohol (IPA). By simply bending the structures along the grooves outlining the outer perimeter, the cut structures were removed as a whole from the bulk metal sheet. Next, individual metal arrays corresponding to the rows in the design were separated from each other by bending and assembled on a piece of tape by hand to achieve the final design. The structure on the piece of tape were then bonded to a magnetic polydimethylsiloxane (PDMS; Sylgard 184, Dow Corning) with pure 10:1 PDMS (Fig. 1C). The magnetic polymer was pre-programmed with the desired magnetisation profile prior to bonding with the structure. The resultant composite structure was then left on a hot plate at 90 °C for 2 h to allow the PDMS to cure. After curing, the tape was removed and the final robot was obtained after the scales in each row and column were separated into separate and distinct scales by bending with tweezers.

### Tumour spheroid experiments

Tumour spheroids were prepared from HT-29 cells. The HT-29 cells were seeded in a Nunclon Sphera-treated, U-shaped-bottom 96-well

microplates (Thermo Scientific) at a density of $2 \times 10^4$ cells per well. The cells were cultured for 72 h in Nunclon spheroid plates in a humidified, 37 °C and 5% $CO_2$ environment. The spheroids' diameter were measured after 72 h incubation (~900 μm). The tumour spheroids were placed on top of an aluminium substrate (1 cm x 1 cm) and then placed at a distance of 5 cm away from the RF coil. The current in the RF heater was set at 621.6 A, which corresponded to a temperature of 60 °C. Depending on the experiment, the RF heater was turned on for 5 min, 10 min, or 15 min. After RF heating, the tumour spheroids were collected and incubated in a humidified 37 °C and 5% $CO_2$ environment for 24 h. The viabilities of tumour spheroids were then measured using the CellTiter-Glo 3D Cell Viability Assay according to the supplier's instructions. The luminescence values were measured in an opaque 96-well plate using a plate reader (BioTek's Synergy 2, Winooski, VT, USA).

## Wireless magnetic soft millirobot fabrication

Polyethylene terephthalate (PET) tape (50650, Tesa) was placed along the sides of a 3 mm acrylic sheet to act as spacers. Next, neodymium-iron-boron (NdFeB) magnetic microparticles (MQP−15−7, Magnequench) were mixed with 10:1 PDMS in a 3:1 weight ratio, stirred for 5 min, degassed for 20 min and poured on the acrylic sheet. Excess mixture was removed with a doctor blade. To cure the mixture, the sample was left on a hot plate at 90 °C for 2 h before it was laser cut (ProtoLaser U3, LPKF Laser & Electronics AG) to the specified dimensions. The final structures were removed from the substrate and wrapped around a 6.4 mm diameter rod. Next, the structures were magnetised in a vibrating sample magnetometer (EZ7, Microsense) with a 1.8 T homogeneous magnetic field for 5 s. To ensure that the robots conformed well to the profile of the rod, a thin layer of water-soluble glue (822095, Pritt) was applied between the robot and the rod. After magnetisation, the robots were detached from the rod by soaking it in DI water. Excess glue on the surface was removed by further rinsing the robot with DI water. The metal scales, which have been prepared separately, were bonded to the robots with 10:1 PDMS (Fig. 1C). The robots, with the scales, were then left to cure on a hot plate at 90 °C for 2 h. A stereomicroscope (ZEISS Stemi 508, Carl Zeiss Microscopy GmbH) was used to guide the process wherever necessary.

## Statistical analysis

All quantitative experimental values were presented as mean ± standard deviation (SD) of the mean.

## Reporting summary

Further information on research design is available in the Nature Portfolio Reporting Summary linked to this article.

## Data availability

All relevant data supporting the key findings of this study are available within the article and its Supplementary Information or from the corresponding author upon reasonable request. Source data are provided with this paper.

## Code availability

Custom codes in MATLAB were used for data processing. The codes are provided in the Supplementary Information accompanying this manuscript.

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

## Acknowledgements

Ex vivo experiments were conducted under the permission: DE08 111 1008 21. Figures 1A, 3A, 4A, and 5A were created with biorender.com. The authors thank Gaurav Gardi, Muhammad Turab Ali Khan and Aniket Pal for discussion, and Devin Sheehan for help with the ex vivo experiments.

## Author contributions

R.H.S., Z.Y., and M.S. proposed the research. R.H.S. and M.S. designed the research with help from all authors. R.H.S. and Z.Y. designed the fabrication protocol. R.H.S. and N.O.D., assisted by A.A., designed and performed the biological and ex vivo demonstrations. R.H.S. and M.A.D., assisted by Z.Y., designed and performed the thermal and mechanical characterisations. R.H.S., M.E.T., A.C.K., and P.E.D. performed the simulations. R.H.S. wrote the manuscript. M.S. supervised the research. All authors analysed the data and discussed the results. All authors edited and commented on the manuscript.

## Funding

This work is funded by the Max Planck Society and European Research Council (ERC) Advanced Grant SoMMoR project with grant number 834531 (M.S.). Open Access funding enabled and organized by Projekt DEAL.

## Competing interests

The authors declare no competing interests.
