## [Peer Review File · Nature Communications]

Reviewer #1 (Remarks to the Author):

This article focuses on the design of an untethered millirobot. Different metal materials are used for heating under radiofrequency (RF) fields, in combination with the assistive effects of external magnetic fields and the demagnetization of magnetic particles, to achieve functionalities such as changing the movement mode of the robot and releasing and heating at targeted locations. The article proposes that the robot's function can be used in the future for drug release in the GI tract and mitigating issues in the body.

There are still many necessary details and content that have not been included, which affects the readability of the article. I have some major comments on this article.

The term "RF" is mentioned in line 59; please modify this place to "radiofrequency" to avoid ambiguity.

Regarding the testing of individual samples in lines 117-119, there is no detailed description of the preparation process and parameters of the samples in the entire text. It is unclear whether the sample is a single layer or multiple layers, what materials were used, and how thick they are.

In the preparation of the soft millirobot, it is possible that the scale could fall off from the PDMS layer during deformation. Is it necessary to improve adhesion through the use of glue or similar substances? If so, will it affect the biocompatibility?

In Fig. 2, the labeling in the chart is unclear. For example, in Fig. 2D, it is unclear what the material is. The description in Fig. 2 does not elaborate. Please clarify similar vague information in the text.

In Fig. 2G, the temperature change finally decreases in the 10-micron thickness, which is inconsistent with other thicknesses. The article provides some explanations, but Fig. S1H does not simulate the 10-micron case. Can you add this case in Fig. S1H and compare the simulation conclusion and experimental data further? Are they consistent? If the two figures are inconsistent, where is the corresponding loophole in the simulation model?

It can be observed from the article and Fig. 4 that the motion mode of the structure changes through demagnetization. However, the significance of this change is not stated in the article. Could the authors provide further explanation of this aspect and its implications for potential applications?

Finally, the text and Fig. 4 also show that specific location release and heating have been realized through demagnetization. However, there is no mention in the article about the limitations and challenges of this method. Can you further discuss this issue in the text and potential solutions?

Reviewer #2 (Remarks to the Author):

Aiming at the development of the magnetic robot that could be movement-controlled via the external magnetic field and capable of localized heat generation for application in the biomedical field, the authors have constructed a pangolin-inspired untethered magnetic robot that can generate heat close to 70 C at the targeted site within 5 cm during ~ 30 seconds. The clever design could definitely improve the performance of this type of devices for the use in clinic. The developers demonstrated the performance (controlled movement and heat generation) of such a device in various Ex Vivo models. Definitely, the capability of this robot capable of active locomotion under a magnetic field as well as removing localized heat generation was demonstrated in vitro and ex vivo.

I have several questions/comments:

1. If I am correct in understanding that the Ex vivo testing was performed on the surface of these selected models, the main question would be how the nature of the environment (e.g. in the intestine or stomach) will influence the movement of this robot. How the viscosity and the content of the biological fluid will influence the robot's movement controlled by magnetic field? What type of fluid is inside the intestine for the experiment depicted in Fig 7?
2. Regarding the safety testing, I am curious if the edges of this robot are not too sharp to damage the tissue? Also, for clinical application, is the intention to remove the robot from the body after the performance and if so, how the complete removal could be guaranteed, as this robot does not seem to be able to biodegrade?
3. In Fig 2, text is too small to read properly even on the screen.
4. It seems that the temperature setting is done in correlation with magnetic field parameters. The biological environment can influence heat generation and the required/expected temperature might be achieved. Perhaps, the heat generation can be evaluated as a function of biological fluids, distance from the magnetic activation source, etc. Perhaps, I missed it but what method is used to record the temperature in Ex vivo models as FLIR detects only surface temperatures?
5. Can be the robot reusable? Even In vitro , does it's performance decreases after multiple applications?
6. Is some heat generated during the application of external magnetic field for remote locomotion?

Reviewer #3 (Remarks to the Author):

This paper proposed a flexible and scalable pangolin-inspired bi-layered design, which is able to achieve significant heating at large distances within a short period of time, allowing users to realize on-demand localized heating in tandem with shape-morphing capabilities. This paper is well organized and well written, and the contents are precisely described. However, the following several issues should be considered and addressed.

-It is necessary to check if the skin depth equation on page 7 of the text is correct
($\delta = \sqrt{\rho / \mu \pi f} = \sqrt{1 / \sigma \mu \pi f}$, $\rho = 1 / \sigma$)

-It is necessary to characterize the heat generated by applying an RF field or an alternating magnetic field to mPDMS.

-In actual clinical application, it is considered that intestinal contents may be caught between the metal plates. So, the locomotion performance of the robot can be deteriorated. I suggest that the related discussion should be included in the manuscript.

-In Fig. 6, I wonder how the hemostatic performance of the robot can be confirmed. To confirm the hemostatic performance of the robot, it is necessary to execute the clotting test. Please consider that the clotting test can be included in the manuscript.

Response Letter

Manuscript ID: NCOMMS-23-03606

We would like to thank the reviewers for their constructive and invaluable comments. We have revised the paper according to these comments, which have further improved the work. **In particular, experimental results and simulations on how the environment impacts heating performance, robot movement and reusability have been added. Additional methodological details have also been provided, along with further discussion on eventual clinical hurdles and limitations.** Comments from the reviewers are appended below in sequence and marked in **blue**, with our responses in **black**. Changes to the manuscript and supplementary material are highlighted in **yellow**.

Reviewer 1

This article focuses on the design of an untethered millirobot. Different metal materials are used for heating under radiofrequency (RF) fields, in combination with the assistive effects of external magnetic fields and the demagnetization of magnetic particles, to achieve functionalities such as changing the movement mode of the robot and releasing and heating at targeted locations. The article proposes that the robot's function can be used in the future for drug release in the GI tract and mitigating issues in the body. There are still many necessary details and content that have not been included, which affects the readability of the article. I have some major comments on this article.

We thank the reviewer for the invaluable comments and feedback. We have addressed all comments point-by-point with clarified descriptions, additional experiments and analyses.

Comment 1:

The term "RF" is mentioned in line 59; please modify this place to "radiofrequency" to avoid ambiguity.

Response:

We thank the reviewer for spotting this typo. We have added the word radio-frequency to the sentence in the revised manuscript.

The sentence starting in line 70 now reads: **“Remote heating can be achieved using alternating magnetic fields or radio-frequency (RF) fields via two mechanisms: Joule heating or hysteresis losses (30).”**

Comment 2:

Regarding the testing of individual samples in lines 117-119, there is no detailed description of the preparation process and parameters of the samples in the entire text. It is unclear whether the sample is a single layer or multiple layers, what materials were used, and how thick they are.

Response:

We thank the reviewer for spotting this lapse.

A new section titled “Sample preparation of samples for comparison of heating performance” (lines 573-589) detailing the fabrication process of individual samples for fig. S1A, together with relevant details such as the dimensions and thicknesses, has been added to the materials and methods section.

“Sample preparation of samples for comparison of heating performance

All thicknesses of the heating component were fixed at 100 μm for a 1 cm^2 square sample. For the mPDMS and eGaIn sample, the liquid metal eGaIn (495425, Sigma-Aldrich) and neodymium-iron-

boron (NdFeB) magnetic microparticles (MMPs, MQP-15-7, Magnequench) were mixed with PDMS (Sylgard 184, Dow Corning), at a weight ratio of 11:110:10:1 (MMPs : eGaIn : PDMS base agent : PDMS curing agent). The mixture was hand-stirred until a consistent emulsion was formed. This was determined visually and typically took 5 min of continuous stirring. The mixture was then degassed for 30 min to remove the trapped air and poured into a mould. Excess mixture was then removed with a doctor blade. The mould was placed on a hot plate at 90 °C for 2 h to allow the composite to cure. For the mPDMS and eGaIn surface sample, the 100 µm-thick liquid metal eGaIn layer was added to the surface of a 11:10:1 (MMP: PDMS base agent: PDMS curing agent) substrate by doctor blading.

mPDMS samples were fabricated by mixing the constituents at a weight ratio of 11:10:1 (MMPs : PDMS base agent : PDMS curing agent). Similarly, the iron (II,III) oxide (637106, Sigma-Aldrich) samples were fabricated using a weight ratio of 11:11:10:1 (MMPs : iron (II, III) oxide, PDMS base agent : PDMS curing agent). After mixing and degassing the mixture for 30 min to remove the trapped air, excess mixture was removed with a doctor blade, and the mould was left on a hot plate at 90 °C for 2 h to allow the composite to cure.”

Comment 3:

In the preparation of the soft millirobot, it is possible that the scale could fall off from the PDMS layer during deformation. Is it necessary to improve adhesion through the use of glue or similar substances? If so, will it affect the biocompatibility?

Response:

We thank the reviewer for the comment.

In the revised manuscript, we performed experiments to measure the force required to tangentially pull an individual scale out (i.e. adhesion of the metal scale to mPDMS). For the overlapping design, the average measured force was 800 mN and was independent of the extent of overlap (fig. S4). This force is significantly higher than the forces which can be generated by a soft magnetic composite, even when ideal conditions are assumed (~ 2 mN) (*I*). As such, we believe that it is unlikely that the scale will fall off from the PDMS during deformation. The following sentence has been added to the main manuscript (Lines 320-322).

“The pull-out force of the aluminium scale on mPDMS was experimentally determined to be approximately 800 mN, implying that the scales are unlikely to fall off during actuation (fig. S4).”

Figure S4. Pull-out force of a 100 μm aluminium scale from mPDMS at different percentage overlaps (n = 5).

Having said that, we do recognise that better alternatives could potentially exist. An example would be Dermabond, which is currently used in clinics for topical wound closures and hence, could potentially provide the necessary adhesion whilst simultaneously offering better biocompatibility. Further studies will have to be conducted to assess the suitability of this adhesive for applications inside the body as the working environment inside the human body varies greatly from the outside. This point has also been added to the discussion (Lines 456-462).

“An example would be Dermabond, which is currently used in clinics for topical wound closures and hence, could potentially provide the necessary adhesion of the scales to mPDMS whilst simultaneously offering better biocompatibility. Further studies will have to be conducted to assess the suitability of this adhesive for applications inside the body as the working environment inside the human body varies greatly from the outside.”

Comment 4:

In Fig. 2, the labelling in the chart is unclear. For example, in Fig. 2D, it is unclear what the material is. The description in Fig. 2 does not elaborate. Please clarify similar vague information in the text.

Response:

We thank the reviewer for the comment.

The figures presented in Fig. 2D are simulated temperatures at $t = 60$ s of a 10 x 10 mm metal scale with varying electrical conductivities and thicknesses. The simulation was conducted in order to isolate and demonstrate the effects of electrical conductivity on the heating performance and do not correspond to any particular material.

However, we acknowledge that the presentation of both experimental and simulated data in the same figure might potentially create confusion for readers. As such, we have included the terms “experimental” and “simulated” in the figure captions to better differentiate the two wherever possible.

Comment 5:

In Fig. 2G, the temperature change finally decreases in the 10-micron thickness, which is inconsistent with other thicknesses. The article provides some explanations, but Fig. S1H does not simulate the 10-micron case. Can you add this case in Fig. S1H and compare the simulation conclusion and experimental data further? Are they consistent? If the two figures are inconsistent, where is the corresponding loophole in the simulation model?

Response:

We thank the reviewer for the comment.

The results for 10 μm were not presented initially because the low thicknesses involved meant that the metal scale had to be modelled with the thin shell approximation rather than as a 3-dimensional (3D) structure in COMSOL. There exists a significant deviation in the maximum temperature from using the approximation, although the trends obtained from using the approximation are similar to those obtained from the 3D model. Since the 3D model was closer to the experimental results, a decision was made to omit the 10 μm results to prevent confusion. However, as the reviewer pointed out, this omission might create more confusion and questions for readers instead. In view of this, we have added the plot back to the manuscript in line 192 and in fig. S1I.

From Figure S1I, we can observe that the peak temperatures occurs between 15 and 20 μm for aluminium. Moreover, although the maximum temperature decreases, the rate of temperature increases as the thickness decreases, as indicated by the increasing slope of the graphs. This is consistent with the experimental measurements presented in Fig. 2G and further demonstrates that our hypothesis and explanation to optimise heating of the scales provided in the manuscript is correct.

Figure S1I. Simulated temperature change of aluminium scale with different thicknesses over time (low thicknesses). Note that the simulated temperatures presented in this figure are lower than those presented previously because the metal scales are modelled with the thin shell approximation in COMSOL.

Comment 6:

It can be observed from the article and Fig. 4 that the motion mode of the structure changes through demagnetization. However, the significance of this change is not stated in the article. Could the authors provide further explanation of this aspect and its implications for potential applications?

Response:

We thank the reviewer for the comment. As there are three sub-questions in the comment, we have broken the comment down and addressed them separately.

However, the significance of this change is not stated in the article.

The significance of changing the magnetic profile is highlighted in the manuscript in lines 343-347, where we describe how the heating plates allow for hitherto unachievable functionalities in soft miniature magnetic robots. The sentences have been highlighted in green in the manuscript.

“Such system could enable in-situ change of the robot’s magnetisation profile, which could enhance the functionalities of these robots. Specifically, miniature magnetic soft robots reported in literature to date have a fixed magnetisation profile which, after deployment, cannot be changed in situ. As such, the response of the robot to an external magnetic field cannot be changed after it is deployed.”

Could the authors provide further explanation of this aspect...

The process of demagnetisation was achieved by heating the magnetic particles above the Curie temperature (lines 333-339). The sentences have been highlighted in green in the manuscript.

“As this was above the Curie temperature of the magnetic particles (MQFP-10-8.5HD-20180, Magnequench) embedded in the soft matrix, the particles shifted to a paramagnetic state and the magnetisation profile previously encoded was lost. After in situ demagnetisation, the robot was observed to be unresponsive to any input magnetic fields, indicating that the robot no longer had any magnetisation profiles (Fig. 4A and movie S2). This was also confirmed separately when the magnetic flux density on the robot was measured (fig. S4).”

... its implications for potential applications?

Lines 350-354 talk about its implications for potential applications in terms of allowing for switching of locomotion modes of a single robot for efficient locomotion over different terrains.

“As such, this demonstration highlights the ability of this method to allow these untethered robots to potentially change the magnetisation on the fly. This further opens up the design space and enhances the performance of untethered miniature robots since a single robot design can now potentially change and adapt its locomotion based on the environment accordingly in situ.”

In the revised manuscript, we have also included two other potential applications (selective actuation and swarm control) in which demagnetisation itself might be useful in a biomedical context (Lines 354-361).

“Moreover, as demonstrated in the movie S2, the demagnetised robot is unresponsive to the actuating fields. As such, through judicious placement of the heating plates, demagnetisation of those parts could enable selective actuation of the robot for specific tasks after the robot has navigated to the target location. Alternatively, this demagnetisation feature could allow for better control of robotic swarms. For instance, untethered robots can be deployed singularly and then deactivated, thereby allowing another robot to be deployed to the same region without the concern that the actuating fields for the second robot would unintentionally actuate the first as well.”

Comment 7:

Finally, the text and Fig. 4 also show that specific location release and heating have been realized through demagnetization. However, there is no mention in the article about the limitations and challenges of this method. Can you further discuss this issue in the text and potential solutions?

Response:

We thank the reviewer for the comment.

We would like to clarify that selective cargo release was enabled by the different heating profiles of the metal scales of different thicknesses. The cargos were attached to the soft magnetic robot with beeswax, which had a melting point of 61 to 65°C. This difference in heating response allowed only the blue cargo to be released, even though the entire soft robot was exposed to the same stimulus (i.e. RF field). Selective cargo release was not achieved through demagnetisation.

We realise that this can potentially be confusing for readers since Fig. 4 shows a demonstration which exploits heating to achieve in situ demagnetisation. As such, the original paragraph (Lines 362-385) has been reworded and an additional sentence has been added to the manuscript lines 365-367 to clarify how selective cargo release was achieved.

“In this variant of the robot, we exploited the different heating rates of the scales to enable selective cargo release.”

In terms of the limitations and challenges, we have also added the following discussion to the manuscript (lines 376-382).

“The main limitation of this robot design lies in the selection of adhesive to secure the cargo to the robot. Although the beeswax used in this demonstration is biocompatible and can be safely ingested, the beeswax re-solidifies upon cooling. This does not pose a problem in this work because we intend to deploy the robot in the GI tract, where the wax can be safely excreted by the human body. However, to further extend the clinical utility of these robots, studies should be conducted to identify adhesives which thermally degrade upon heating into compounds which can be safely excreted or absorbed by the body.”

Reviewer 2

Aiming at the development of the magnetic robot that could be movement-controlled via the external magnetic field and capable of localized heat generation for application in the biomedical field, the authors have constructed a pangolin-inspired untethered magnetic robot that can generate heat close to 70 C at the targeted site within 5 cm during ~ 30 seconds. The clever design could definitely improve the performance of this type of devices for the use in clinic. The developers demonstrated the performance (controlled movement and heat generation) of such a device in various Ex Vivo models. Definitely, the capability of this robot capable of active locomotion under a magnetic field as well as removing localized heat generation was demonstrated in vitro and ex vivo.

We thank the reviewer for the recognition and support of our work. We have addressed all following comments point-by-point with clarified descriptions, additional experiments and analyses.

Comment 8:

If I am correct in understanding that the Ex vivo testing was performed on the surface of these selected models, the main question would be how the nature of the environment (e.g. in the intestine or stomach) will influence the movement of this robot. How the viscosity and the content of the biological fluid will influence the robot’s movement controlled by magnetic field? What type of fluid is inside the intestine for the experiment depicted in Fig 7?

Response:

We thank the reviewer for the comment.

The fluid used in Fig. 7 was DI water to facilitate imaging with ultrasound. The following sentence has been added to the caption for Fig. 7 for clarity.

“Small intestine is filled with DI water.”

The pangolin-inspired overlapping design introduced in this work utilises rolling for locomotion. In terms of how the viscosity and the contents of the biological fluid would influence rolling, we refer to a publication (2), in which a rolling wheel (no slip) was found to be a good approximation for the robot’s rolling at low frequencies (< 10 Hz). As such, the equation of motion for the rolling robot can be expressed as:

$$F_T - F_A - F_R = M\ddot{x} \quad (1)$$

where F_T is the traction force, F_A is the aerodynamic drag, F_R is the rolling resistance, M is the mass of the robot and \ddot{x} is the linear acceleration.

At steady state, $\ddot{x} = 0$ and Eq. 1 reduces to:

$$F_T = F_A + F_R \quad (2)$$

Next, we consider each term in Eq. 2 separately.

F_T can be re-expressed as:

$$F_T = \frac{\tau_{Roll}}{r_{eff}}$$

where τ_{Roll} is the magnetic torque applied and r_{eff} is the effective radius.

F_A can be re-expressed as:

$$F_A = \frac{1}{2} \rho AV^2 C_D$$

where ρ is the density of the fluid, A is the reference area, V is the velocity and C_D is the drag coefficient. In this work, A is the frontal area and can be taken to be the following:

$$A = 2r_{eff} \cdot b$$

where b is the width of the robot.

On the other hand, C_D is sensitive to changes in Reynold's number (Re) and can vary by several orders of magnitude. To illustrate this, we consider the 20 x 10 x 0.2 mm robot used in this work. Assuming the robot's velocity to be 0.05 m/s, a characteristic length of 0.01 m, and the density of fluid to be 1000 kg/m³ – the major component of most biological fluids is water and remains fairly constant (3), sweeping the dynamic viscosity from 10⁻³ to 10² (4) would result in a Re in the range of 10⁻³ to 10². Consequently, C_D can vary anywhere between 10⁰ to 10² (5).

For the rolling resistance, F_R , hysteresis losses constitute the bulk of it. For a wheel, these losses are dependent on the normal load, temperature, speed and contact friction (i.e. friction between the metal scale and tissue surface) (6).

Taken together, this implies that the contact friction and viscosity are the dominant environmental properties affecting magnetic actuation. When the viscosity is increased, the resultant increase in Re and C_D , means that F_A would increase correspondingly. This results in a lower \ddot{x} and speed. Similarly, when the contact friction is increased, a lower speed would be achieved given the same inputs (i.e. actuating magnetic (B) field and magnetisation, M, of the robot).

The extent to which rolling is affected highly depends on how these parameters change and requires further studies for a number of reasons. Firstly, biological fluids are typically non-Newtonian and exhibit shear thinning behaviours (4). Moreover, the above analysis only holds true when the entire robot is submerged in the fluid. In certain scenarios, such as those presented in this work, the robot might only be partially submerged in the fluid (e.g. only the scales). Lastly, the scales in this work could potentially even enhance locomotion. In this regard, similar to how snow chains help wheels to grip the road and prevent skidding, these scales could also help the robot dig into a layer coated with mucus and therefore prevent the robot from slipping.

A new section titled “**Environmental impact on locomotion**” has been added to the SI. The following sentence has also been added to the manuscript.

(Lines 416-418)

Results from the initial biocompatibility tests for the long-term use of aluminium inside the human body and how locomotion will be affected by the environment are presented in *SI Appendix, Initial Biocompatibility Studies* and *SI Appendix, Environmental impact on locomotion*, respectively.

(Lines 438-440)

“For successful translation, future studies should concentrate on four technical aspects on top of application specific issues, such as how residual intestinal content might still be present after flushing of the GI tract and how it may interfere with locomotion.”

Comment 9:

Regarding the safety testing, I am curious if the edges of this robot are not too sharp to damage the tissue? Also, for clinical application, is the intention to remove the robot from the body after the performance and if so, how the complete removal could be guaranteed, as this robot does not seem to be able to biodegrade?

Response:

We thank the reviewer for the comments.

As pointed out by the reviewer, tissue penetration could be a potential cause for concern in the overlapping robot variant because the metal plates would jut out of the structure when the robot is deformed. Contact with the metal scales especially around the corners could then potentially cause damage to the underlying tissue layers. Even so, in terms of deploying the robot inside the human body, there are two factors which, in our opinion, make tissue penetration and hence, damage to the underlying tissue layer highly unlikely.

Firstly, internal organs and cavities are lined with mucus, a protective viscoelastic membrane. In the gastrointestinal (GI) tract, where the proof of concept demonstrations were performed, the thickness of this mucus layer ranges from 0.4 to 0.8 mm depending on the location along the GI tract (7). It is important to note that these in vivo values were obtained in rats and as such, it is likely that these values (i.e. thickness of mucus layer) would be much larger in humans. Consequently, this implies that tissue penetration might only occur when larger scales at low overlaps are used (e.g. 25% overlap of 5 mm scales resulting in a 3.75 mm overhang). The flexibility of adopting the overlapping pangolin inspired design becomes more apparent here; it offers users with a possibility to circumvent this issue without a significant deterioration in heating performance. Specifically, the drop in heating performance due to the use of a smaller scale to prevent tissue penetration can be offset by using smaller scales with higher degrees of overlap (Fig. 2H and I).

Secondly, the soft body provides used in the fabrication of the robot inherently provides the necessary compliance and safety to prevent tissue penetration. Specifically, tissue penetration requires an average force of 0.1-1 N (8-10), which is at least 1000 times higher than the forces which can be generated by a soft magnetic composite, even when ideal conditions are assumed (1, 11). As an additional layer of safety, the sharp edges can be rounded with fillets. This would remove sharpness (and hence the stress concentrations required for tissue puncture) from the corners of the scales without unduly affecting the heating performance – eddy currents do not flow in the regions close to the corners. We demonstrate this understanding of the system from the results presented in fig. S11, where we observed that the heating performance (i.e. maximum temperature and rise time) achieved from the scaled design with and without a 0.2 mm fillet were similar. This discussion has been added to the manuscript (Lines 419-421).

“Moreover, the safety of deploying these robots inside the human body can be further enhanced by adding fillets to the scales to reduce the likelihood of tissue puncture, without adversely affecting the overall heating performance (fig. S11).”

Figure S11. Heating performance of 16 non-overlapping 2.5 mm square 100 μ m-thick aluminium scales with 0.2 mm fillets as compared to another with identical dimensions without fillets (n = 6).

Regarding the point about complete removal, the reviewer is correct in pointing out that the robot, in its current form, is non-biodegradable. For this reason, we limited the applications to those in the GI tract, where the robot could be safely removed/excreted; it is smaller in volume than an existing FDA approved non-biodegradable oral drug delivery capsule (10). Moreover, the active and controllable magnetic locomotion would offer more flexibility and options in helping to non-invasively steer the robot out of tortuous regions in the GI tract as opposed to their passive counterparts (i.e. capsules). Despite this, we recognise for future deployment out of the GI tract, the use of biocompatible and biodegradable materials such as FePt and hydrogels as substitutes for mPDMS should be considered. The relevant discussion has been added to the manuscript (Lines 461-462).

“Biocompatible and biodegradable materials such as FePt and hydrogels as substitutes for mPDMS should also be considered.”

Comment 10:

In Fig 2, text is too small to read properly even on the screen.

Response:

We thank the reviewer for pointing this out.

We have enlarged all figures in both the main and SI in the revised version.

Comment 11:

It seems that the temperature setting is done in correlation with magnetic field parameters. The biological environment can influence heat generation and the required/expected temperature might be achieved. Perhaps, the heat generation can be evaluated as a function of biological fluids, distance from the magnetic activation source, etc. Perhaps, I missed it but what method is used to record the temperature in Ex vivo models as FLIR detects only surface temperatures?

Response:

We thank the reviewer for the comment.

In terms of evaluating the heat generation as a function of biological fluids and distance from magnetic activation source (i.e. RF coil), we have supplemented the original figures with additional figures from simulations in the SI. The updated results and discussion are provided in SI fig. S7 and Lines 392-400, respectively.

“Moreover, to account for the highly heterogeneous environments inside the body, simulations were performed to determine the heating performance of the scales across different distances and convective heat transfer coefficients. This would provide insights on the heating performance of the robot under more realistic conditions. Based on the additional simulations performed (fig. S7), we observed that for a similar decrease in heating performance (i.e. final temperature of the metal scale after 60 s of RF exposure), the distance from the coil had to be increased by 3 times while the convective heat transfer coefficient had to be increased by 1000 times for a given RF input. This implies that the heating performance is more sensitive to changes in the magnetic flux (i.e. distance) as opposed to changes in heat losses (i.e. environmental conditions).”

Regarding the second point, the reviewer is correct in pointing out that FLIR only detects the surface temperature. The desired temperatures on the robot in the demonstrations were achieved by performing a calibration before the actual experiment. In this regard, the robot to be used in the demonstration was placed on a glass slide and exposed to the RF field with FLIR keeping track of the temperature. The time taken to reach a temperature 10% higher than the desired temperature, to compensate for additional thermal losses, was then recorded. This time was then used during the actual demonstrations. This information has also been added to the updated manuscript (Lines 536-541).

“The desired temperatures on the robot in the demonstrations were achieved by performing a calibration before the actual experiment. In this regard, the robot to be used in the demonstration was placed on a glass slide and exposed to the RF field while the temperature was tracked with FLIR. The time taken to reach a temperature 10% higher than the desired temperature, to compensate for additional thermal losses, was then recorded. This time was then used during the actual demonstrations.”

Figure S7. Simulated temperatures of a 100 μm aluminium scale at $t = 60$ s at different convective heat transfer coefficients and distances from the RF coil.

Comment 12:

- Can be the robot reusable? Even In vitro , does it's performance decreases after multiple applications?

Response:

We have performed additional experiments showing the repeatability and reliability of the device over 30 heating and cooling cycles. From the results presented in fig. S2G and H, it is observed that there was no noticeable decrease in the heating performance over 30 cycles. Over the 30 cycles, the final temperature of the scaled structure after 1 min of RF exposure fell between 5% of the average value as demarcated by the red region. Furthermore, we autoclaved the same sample and showed that there was only a 5% decrease in heating performance. This information has been included in the manuscript (Lines 220-222).

“The heat produced by the scales is also highly reliable and repeatable, with the maximum temperature reached varying by less than 5% over 30 cycles. Even after it was autoclaved, the heating performance degraded by less than 5% (fig. S2G and H).”

Figure S2. (G) Temperature change of 16 non-overlapping 2.5 mm 100 μm -thick aluminium scales over 30 heating and cooling cycles. Region demarcated in red indicates the temperatures that fall within 5% of the average maximum value. (H) Temperature change of 16 non-overlapping 2.5 mm 100 μm -thick aluminium scales before and after autoclaving ($n = 5$).

Comment 13:

- Is some heat generated during the application of external magnetic field for remote locomotion?

Response:

No noticeable heating was observed during the application of an external magnetic field for locomotion. Joule heating resulting from the generation of eddy currents can be described and modelled with Faraday's Law.

$$\nabla \times \mathbf{E} = -\frac{\partial \mathbf{B}}{\partial t}$$

As observed from the equation, the electric field produced, and hence the current generated in the metal scale, is dependent on the rate of change of magnetic flux through the area enclosed by the scale. This implies that the heating performance is dependent on the frequency and the amplitude of the magnetic field.

In this work, the frequency of magnetic fields used for locomotion was at most 3 Hz while the frequency of the magnetic fields used for heating was at 338 kHz. Furthermore, the magnitude of the magnetic fields used for locomotion and heating were 65 mT and 0.5 mT, respectively. Taken together, this

implies that the electric field which will be generated from the magnetic fields used for locomotion would be 10^3 times smaller than that generated using the RF fields in an ohmic material.

As such, although eddy currents would still be generated since Faraday's law applies in both cases, the induced currents and any resultant increase in temperature arising from the magnetic fields for locomotion would be too minute for any meaningful application.

Reviewer 3

This paper proposed a flexible and scalable pangolin-inspired bi-layered design, which is able to achieve significant heating at large distances within a short period of time, allowing users to realize on-demand localized heating in tandem with shape-morphing capabilities. This paper is well organized and well written, and the contents are precisely described. However, the following several issues should be considered and addressed.

We thank the reviewer for the acknowledgement and support of our work. We have addressed all following comments point-by-point with clarified descriptions, additional experiments and analyses.

Comment 14:

It is necessary to check if the skin depth equation on page 7 of the text is correct ($\delta = \sqrt{\rho / \mu \pi f} = \sqrt{1 / \sigma \mu \pi f}$, $\rho = 1 / \sigma$)

Response:

We thank the reviewer for spotting this typo. We have corrected the equation to the following in the revised manuscript:

$$\delta_{xy} = \sqrt{\frac{1}{\sigma \mu \pi f}}$$

Comment 15:

It is necessary to characterize the heat generated by applying an RF field or an alternating magnetic field to mPDMS.

Response:

We thank the reviewer for the comment.

The contributions of mPDMS to the heat generated are presented in fig. S1A and the main text (lines 119-121). The sentence has been highlighted in green in the manuscript. Specifically, for mPDMS, the heat generated when exposed to an RF field would be predominantly from magnetic hysteresis, which exhibits poor performance over large distances (i.e. 5 cm). As such, there will be no noticeable increase in temperature when the sample is exposed to the RF field (fig. S1A).

Having said that, we realise that the overlap of three graphs (i.e. mPDMS, iron (II, III) oxide and eGaIn droplets) makes it hard for readers to distinguish between them and could cause confusion. As such, the following description has been added to the caption for fig. S1 for clarity.

“As the samples made from mPDMS, mPDMS and eGaIn, and iron (II, III) oxide do not record any temperature change, the graphs of these samples overlap one another.”

Comment 16:

In actual clinical application, it is considered that intestinal contents may be caught between the metal plates. So, the locomotion performance of the robot can be deteriorated. I suggest that the related discussion should be included in the manuscript.

Response:

We thank the reviewer for the comment.

Similar to how endoscopy or colonoscopy is currently performed, we envision that the patient would have to fast and have the GI tract flushed, before any procedure can be performed with the robot. This would greatly reduce the amount of intestinal content present. Despite this, we acknowledge that some residual intestinal content might still be present and may interfere with locomotion. In view of this, we have added the following sentences to the discussion (Lines 438-440).

“For successful translation, future studies should concentrate on four technical aspects on top of application specific issues, such as how residual intestinal content might still be present after flushing of the GI tract and how it may interfere with locomotion.”

Moreover, a new section titled “*Environmental impact on locomotion*” has been added to the SI, in which an analytical analysis on locomotion was conducted. From the analysis, it was observed that viscosity and contact friction are the dominant properties affecting magnetic actuation. The following sentence has also been added to the manuscript.

(Lines 416-418)

Results from the initial biocompatibility tests for the long-term use of aluminium inside the human body and how locomotion will be affected by the environment are presented in *SI Appendix, Initial Biocompatibility Studies* and *SI Appendix, Environmental impact on locomotion*, respectively.

Comment 17:

In Fig. 6, I wonder how the hemostatic performance of the robot can be confirmed. To confirm the hemostatic performance of the robot, it is necessary to execute the clotting test. Please consider that the clotting test can be included in the manuscript.

Response:

We thank the reviewer for the comment.

To the best of our knowledge, clotting tests such as those performed in hospitals are performed either directly with assays or indirectly through measuring the clotting time or clot solubility. These tests typically test for specific coagulation factors/proteins/plasma (e.g. factor VII, VIII, IX, X, XIII, fibrin D-dimer etc.) and as such, can be broadly classified as chemical tests. These tests predominantly test for patient’s own hemostatic response (i.e. coagulation cascade).

On the other hand, bleeding was mitigated in this work (movie S5 and Fig. 6) using a physical method (i.e. heating). This method more closely resembles the cauterisation procedure in medical practice today than what clotting tests typically test for. Even if the above-mentioned tests were performed, the results cannot be used to attribute the cause of hemostasis to heat application. Moreover, visual methods are being used to determine clotting in hospitals today. For instance, apart from blood tests, ultrasound, X-rays and MRI (i.e. visual methods) are also used to diagnose deep-vein thrombosis (12, 13). As such, in the demonstration presented in movie S5 and Fig. 6, a visual method to determine if bleeding had indeed been stopped was used. Specifically, after the application of heat to the bleeding site, no noticeable increase in the volume of blood was observed (from 1:20 onwards in movie S5).

References

1. R. H. Soon, Z. Ren, W. Hu, U. Bozuyuk, E. Yildiz, M. Li, M. Sitti, On-demand anchoring of wireless soft miniature robots on soft surfaces. *Proc. Natl. Acad. Sci.* **119**, 1–11 (2022).
2. W. Hu, G. Z. Lum, M. Mastrangeli, M. Sitti, Small-scale soft-bodied robot with multimodal locomotion. *Nature.* **554**, 81–85 (2018).
3. P. A. Hasgall, F. Di Gennaro, C. Baumgartner, E. Neufeld, B. Lloyd, M. C. Gosselin, D. Payne, A. Klingeböck, N. Kuster, IT'IS Database for Thermal and Electromagnetic Parameters of Biological Tissues (2022), , doi:10.13099/VIP21000-04-1.
4. S. K. . Lai, Y.-Y. Wang, D. Wirtz, J. Hanes, Micro- and macrorheology of mucus. *Adv. Drug Deliv. Rev.* **61**, 86–100 (2009).
5. R. K. Finn, Determination of the Drag on a Cylinder at Low Reynolds Numbers. *J. Appl. Phys.* **24**, 771–773 (1953).
6. J. C. Páscoa, F. P. Brójo, F. C. Santos, P. O. Fael, An innovative experimental on-road testing method and its demonstration on a prototype vehicle. *J. Mech. Sci. Technol.* **26**, 1663–1670 (2012).
7. M. E. V. Johansson, J. M. H. Larsson, G. C. Hansson, The two mucus layers of colon are organized by the MUC2 mucin, whereas the outer layer is a legislator of host–microbial interactions. *Proc. Natl. Acad. Sci.* **108**, 4659–4665 (2011).
8. A. M. Okamura, C. Simone, M. D. O’Leary, Force Modeling for Needle Insertion Into Soft Tissue. *IEEE Trans. Biomed. Eng.* **51**, 1707–1716 (2004).
9. A. Abramson, E. Caffarel-Salvador, V. Soares, D. Minahan, R. Y. Tian, X. Lu, D. Dellal, Y. Gao, S. Kim, J. Wainer, J. Collins, S. Tamang, A. Hayward, T. Yoshitake, H.-C. Lee, J. Fujimoto, J. Fels, M. R. Frederiksen, U. Rahbek, N. Roxhed, R. Langer, G. Traverso, A luminal unfolding microneedle injector for oral delivery of macromolecules. *Nat. Med.* **25**, 1512–1518 (2019).
10. A. Abramson, E. Caffarel-Salvador, M. Khang, D. Dellal, D. Silverstein, Y. Gao, M. R. Frederiksen, A. Vegge, F. Hubálek, J. J. Water, A. V. Friderichsen, J. Fels, R. K. Kirk, C. Cleveland, J. Collins, S. Tamang, A. Hayward, T. Landh, S. T. Buckley, N. Roxhed, U. Rahbek, R. Langer, G. Traverso, An ingestible self-orienting system for oral delivery of macromolecules. *Science (80-)*. **363**, 611–615 (2019).
11. M. Li, Y. Tang, R. H. Soon, B. Dong, W. Hu, M. Sitti, Miniature coiled artificial muscle for wireless soft medical devices. *Sci. Adv.* **8**, eabm5616 (2022).
12. W. Lim, G. Le Gal, S. M. Bates, M. Righini, L. B. Haramati, E. Lang, J. A. Kline, S. Chasteen, M. Snyder, P. Patel, M. Bhatt, P. Patel, C. Braun, H. Begum, W. Wiercioch, H. J. Schünemann, R. A. Mustafa, American Society of Hematology 2018 guidelines for management of venous thromboembolism: diagnosis of venous thromboembolism. *Blood Adv.* **2**, 3226–3256 (2018).
13. J. Hirsh, R. D. Hull, G. E. Raskob, Clinical features and diagnosis of venous thrombosis. *J. Am. Coll. Cardiol.* **8**, 114B-127B (1986).

Reviewer #1 (Remarks to the Author):

The authors have addressed my comments.

Reviewer #2 (Remarks to the Author):

The authors addressed all my concerns. I recommend publishing this manuscript in its updated form.

Reviewer #3 (Remarks to the Author):

All comments were properly addressed in the reply sheet and the revised manuscript.

So, I consider that this manuscript can be accepted.